# Ellipsoidal Optimal Recovery: A Minimax Approach to Robust Counterfactual Estimation

## Abstract

Consider the problem of quantifying the causal effects of an intervention to determine whether the intervention achieved desired outcomes. Researchers address this problem using statistical, machine learning, or signal processing techniques that have limitations of high bias or need of expert knowledge. We present a new minimax geometric approach called *ellipsoidal optimal recovery (EOpR)* for estimating the unobservable outcome of a treatment unit. It is an approximation-theoretic technique that recovers unknown observations given a learned signal/principal vector and a set of known observations. The significance of our approach is that it improves pre-treatment fit and mitigates bias of the post-treatment estimate relative to other methods in causal inference. Beyond recovery of the unit of interest, an advantage of EOpR is that it produces worst-case limits over the estimates produced. We assess our approach on synthetically-generated data, on standard datasets commonly used in the econometrics (synthetic control) literature, and in the context of the COVID-19 pandemic, showing better performance than baseline techniques.

**Keywords:** causal inference, optimal recovery, synthetic control

## 1 Introduction

An important part of policy evaluation is to estimate the effects of an implemented policy, so as to know whether it achieved its goals. Estimating effects further yields inferences of causal relationships between interventions and their outcomes. Quantifying the effect of a treatment has been of interest not only in policymaking but also across different domains in health sciences, social sciences, and engineering. Typically, the effect is measured by looking at the difference between outcomes before and after an intervention of a treated unit. However, for a given object (e.g., geographical region) at a given time, only one of the outcomes is observed and not both. Thus, we aim to recover and estimate the outcome that was not observed.

Several causal inference methods have been developed for observational studies to estimate the unobserved outcomes for a given intervention. For example *synthetic control* (SC) by Abadie (2021) constructs a weighted average of control units to act as a synthetic control unit to compare with the treated unit. Recently, there has been a growing literature that approaches causal inference from a matrix completion perspective. Proposals include approximating the control unit matrix using nuclear-norm minimization (Athey et al., 2021), using singular value decomposition (Amjad et al., 2018), or by finding nearest neighbors (Agarwal et al., 2021) for missing entries of a matrix to best match control units and the treated unit of interest.

A main limitation in previous work is that when there is insufficient data, especially data from only a short period of time, methods are unable to recover the true estimate (Amjad et al., 2019). Further, the insufficiency or the low quality of data tend to create a poor pre-treatment fit which is a main source of bias in estimates in SC (Abadie, 2021).

**Our Contributions.** This work has three main contributions: algorithmic, theoretical, and empirical.

*Algorithmic.* Our first contribution is to develop an approximation-theoretic approach called *ellipsoidal optimal recovery* (EOpR) that minimizes the worst-case error to optimally recover the unobserved outcome. This also addresses the problem of insufficiency of data, specifically with short pre-intervention period.

Building on methods from signal processing, we adapt the *optimal recovery* algorithm (Muresan & Parks, 2004) to recover missing outcomes. Unlike statistical approaches (Abadie & Gardeazabal, 2003; Abadie et al., 2010) or low-rank matrix decomposition approaches (Amjad et al., 2018; 2019; Athey et al., 2021) which minimize the average error, our approach minimizes the maximum error over the known samples, which is intended to make EOpR a robust algorithm, among other properties. It has previously been proven that minimizing the maximum error produces robust estimators with small bias (Kassam & Poor, 1985; Muresan & Parks, 2005).

**Why Ellipsoids?** We posit that phenomena under study have signals that belong to some particular ellipsoidal signal class, i.e. a quadratic function class (a particular type of convex function class). This is a common assumption in nonparametric econometrics in which signal classes are constructed to characterize bounded smoothness or shape restrictions (Armstrong & Kolesár, 2018; Cheng et al., 1997). It is a valid assumption since nearly all real-world societal signals are smooth. Moreover in many real-world scenarios, data is also naturally bounded, e.g. income data, test scores, gross domestic product (GDP), and others. Such quadratic signal class assumptions also abound in numerous applications in science and engineering (Lorentz et al., 1996). Therefore, convex functions are better to adapt to the bounded data structure and produce estimates that are accurate with minimal bias (Cheng et al., 1997).

Let us separately note that Armstrong & Kolesár (2018) show that assuming convex function classes (such as ellipsoids) enables optimal inference in linear regression. Indeed, the degree of smoothness of a function directly governs how well it is approximated, e.g. using low-degree polynomial reconstruction (Lorentz et al., 1996). Further, Armstrong & Kolesár (2018) derive minimax bounds and confidence intervals over convex classes under shape or smoothness constraints. This jives with our motivations to consider ellipsoidal classes and obtain minimax estimates.

In our setting, the ellipsoidal signal class captures the underlying geometric structure and dependencies in the data, enabling robust extrapolation of the treated unit even under a short pre-treatment period or noisy conditions. Note that our method does not aim to be universally applicable, but rather provides a minimax optimal solution for ellipsoidal classes, which is suitable for numerous practical scenarios especially in the noisy and low-sample regime.

***Theoretical.*** Our second contribution is the derivation of desired properties given the geometrical structure underlying EOpR. Taking the ellipsoidal assumption allows us to develop several useful properties: (a) a quadratic closed-form solution, (b) optimization problem that is min-max by the definition of the circle/ellipsoid, as we see throughout the paper, and (c) solution that is bounded given the boundary conditions of the ellipsoid, such that we prove the consistency of the estimator and the derivation of worst-case estimates of outcomes given the geometrical properties of the algorithm.

All such properties are novel to the causal inference literature and yet are desirable. The synthetic control literature has mainly considered regression-style solutions with a minimization objective only, and has not considered min-max objectives with worst-case estimation bounds.

***Empirical.*** Our third contribution is the extensive evaluation of our algorithm, comparing to numerous baselines. We conduct two sets of experiments: (a) on synthetically generated data following Amjad et al. (2018); (b) on existing case studies from real-world datasets used by Abadie & Gardeazabal (2003); Abadie et al. (2010); Abadie (2021) and an additional real-world COVID-19 experiment that we developed. Synthetic data helps validate the efficacy of our method, since it is impossible to simultaneously observe the treated unit and its counterfactual. Our algorithm outperforms baseline methods especially under varying lengths of pre-treatment periods.

**Organization of the paper.** Section 2 reviews some aspects of causal inference and related work. Section 3 formally states our problem setup. Section 4 describes our estimation strategy in the context of comparative studies data. Section 5 supports the efficacy of our approach via artificial and empirical experiments.

## 2    Related Work

To fix concepts and common terms, we briefly overview causal inference and discuss related work.

Causal analysis takes a step beyond standard statistical analysis, inferring beliefs under changing conditions to uncover causal relationships among variables (Pearl, 2009). Several frameworks have been proposed to tackle causality analysis, such as structural models (Pearl, 2010) and the potential outcome framework (Rubin, 1974), which we focus on here.

## 2.1 Potential Outcomes Framework

This framework assumes effects are tied to a treatment or an intervention. To reveal the causal effects of an intervention, Rubin (1974) proposed to measure the difference of two potential outcomes. The outcome for a unit without being exposed to an intervention and the outcome after an intervention is applied. So, the causal effect is the difference between the two outcomes. However, in real applications, we can never observe both outcomes for the same unit under the same conditions, as only one of the two will take place at a given time. Therefore, one of the potential outcomes will always be *missing*, and the core objective of the framework is to estimate the missing outcome.

Let us introduce the main terms used in the potential outcomes literature, which are used throughout. A *unit* is the atomic object in the framework, which can be any object, whether it is a patient or a city. A *treatment*[1] is the action applied on a unit to change its state, whether administering medicine or enforcing a lockdown order. Treatment is usually thought of as binary, so one group receives the treatment (the *treated* group) and the other does not (the *control* group). *Panel data* is another word for cross-sectional time-series data and comparative studies.

## 2.2 Related Causal Inference Methods

Synthetic control (SC) proposes a particular way to measure the missing observable potential outcome to estimate causal effects (Abadie et al., 2010). Instead of using a single control unit or a simple average of a set of control units, SC creates a *synthetic* unit to act as a control group by selecting appropriate weights for selected control units. The choice of weights should result in a synthetic control unit that best resembles the pre-intervention values of the treated unit.

Therefore, SC is subject to the curse of dimensionality, in which the probability of exact weight matching vanishes as the number of time periods increases (Ferman & Pinto, 2021; Ben-Michael et al., 2021). The *demeaned SC* (DSC) aims to relax SC constraints on weights to allow for a good pre-treatment fit when the length of pre-intervention is very large (Ferman & Pinto, 2021). Similarly, the panel data approach improves performance with increasing pre-intervention periods, using unconstrained regressions relaxing the SC assumptions (Wan et al., 2018). On the other hand, for shorter pre-intervention periods, SC weights fail to reproduce the trajectory of the treated unit (Abadie, 2021) or overfit to idiosyncratic errors (Sun et al., 2024).

Further, SC tends to heavily depend on the expert selection of control units; therefore, estimates are biased by noisy control units if they exist. *Robust* SC (RSC) aims to enhance SC by denoising the set of control units using a latent variable model (Amjad et al., 2018). The observable outcomes of the control group are obtained by a low-rank approximation.

Recent work on SC finds that adding multiple outcomes metrics, e.g., GDP and education level, improves inference (Amjad et al., 2019). Instead of a separate SC model for each outcome, Sun et al. (2024) propose having a common set of weights across multiple outcome series, either by balancing a vector of all outcomes or an average of them. Though this approach is interesting, we do not consider it in the present work.

Recent advances in balancing weights explore diverse strategies to optimize covariate balance while controlling estimation error. Bruns-Smith & Feller (2022) considers balancing weights with a simple convex loss to find the minimum dispersion weights that constrain the worst-case bias between controls over an outcome. While Kallus (2020) balance weights by reproducing kernel Hilbert space (RKHS) by minimizing worst-case conditional mean squared error under smoothness assumptions. This approach generalizes and subsumes

---

[1]The terms *treatment* and *intervention* are used interchangeably.

existing exact matching methods, and propensity score weighting. Notably, Bruns-Smith et al. (2025) show that weighting methods can simultaneously achieve covariate balance and efficient outcome modeling.

Another variation of balancing weights methods is meant to deal with a dynamic data collection setting, e.g., personalized recommendation systems, by updating the principal components as new data arrives (Agarwal et al., 2023), or by solving the model in a reinforcement learning fashion (Dwivedi et al., 2024).

Here, we consider a linear panel data setting, with concentration on a single metric outcome and static data. For this problem setting, none of the recently-proposed methods address the problem of emerging bias and overfitting in short pre-intervention periods.

## 3  Problem Formulation

Consider panel data that is a collection of time series with respect to an aggregated metric of interest (e.g., country GDP). The data includes $N$ units observed over $T$ periods of time. Let $T_0$ be the intervention time, which splits the time period into a pre-intervention period over $1 \leq T_0 < T$ and a post-intervention period with length $T_1 = T - T_0$. For the treated unit of interest, we fix $i = 1$ ($i \in \{1, \ldots, N\}$) for time $t \leq T$, hence, let $s_{1t} \in \mathbb{R}^T$, or for brevity, we also use $s_1$. The remaining units $i = 2, \ldots, N$ are the controls that are not affected by the intervention. Let $\boldsymbol{S} \in \mathbb{R}^{N-1 \times T}$ be the control units matrix.

Let the outcomes of the control and treated units follow a factor model, a common model in the econometrics literature (Abadie, 2021). Let $x_{it}$ denote an aggregated metric for a unit $i$ at time $t$. In the absence of covariates and unobserved outcomes, the factor model is the following:

$$x_{it} = s_{it} + \epsilon_{it}. \tag{1}$$

Following the literature in data imputation (Athey et al., 2021; Amjad et al., 2018) and optimal recovery for missing values (Muresan & Parks, 2004), we consider $s_{it} = f(\eta_i \psi_t)$, where $\eta_i \in \mathbb{R}^N$ and $\psi_t \in \mathbb{R}^T$ are the latent features that capture the unit and time specifications respectively for observed outcomes. The function $f(\cdot)$ captures independent random zero-mean noise, $\epsilon_{it}$, with variance $Var(\epsilon_{1j}) \leq \sigma^2$. Equation 1 can be rewritten in matrix form as $\mathbf{X} = \mathbf{S} + \mathcal{E}$.

To distinguish pre- and post-intervention periods, let $\boldsymbol{A} = [\boldsymbol{A}^-, \boldsymbol{A}^+]$, where $\boldsymbol{A}^- = \{a_{it}\}_{2 \leq i \leq N, t \leq T_0}$, and $\boldsymbol{A}^+ = \{a_{it}\}_{2 \leq i \leq N, T_0 < t \leq T}$. Vectors are defined in the same manner, i.e. $a_i = [a_i^+, a_i^-]$. The inverse of $\boldsymbol{A}$ is $\boldsymbol{A}^{-1}$. The Moore-Penrose pseudo-inverse of $\boldsymbol{A}$ is $\boldsymbol{A}^\dagger$. The transpose of $\boldsymbol{A}$ is $\boldsymbol{A}^\top$. The $\boldsymbol{A}$-norm of a vector $u$, denoted $||u||_A$, is the value of $u^\top \boldsymbol{A} u$. *(Note that the subscript $t$ is used to index a specific time period of a vector or matrix, while the negative sign $(-)$ is used as superscript to reflect the range for all $t \leq T_0$, used to index pre-intervention period of matrices and vectors, similarly the positive sign $(+)$ reflects the post-intervention range for all $t > T_0$).*

The same notational convention holds for $\boldsymbol{S}$ and $\boldsymbol{X}$. We keep $\boldsymbol{A}$ as a generic variable. We use $\boldsymbol{S}$ to reflect a theoretical setting to develop the basic geometric algorithm and its properties. We consider $\boldsymbol{X}$ to reflect a noisy setting, i.e. the latent factor model in equation 1, for actual deployment in simulations and real-world experiments.

Given the noisy observations $\boldsymbol{X}$, and the partially observed treated unit $s_1^-$ (in pre-intervention), our algorithm approximates $s_1^+$, the missing part of the treated unit vector (in post-intervention).

Notably, we use the term latent factors to refer to low-dimensional factors that generate the time series trajectories of both treated and control units. This aligns with standard assumptions in the synthetic control literature and factor modeling (Athey, et al., 2021;Abadie, et al., 2010; Amjad et al., 2018). The latent space is assumed to influence the outcome process, unlike confounding factors that are assumed to influence the treatment assignment process. We defer studying confounding factors to future work.

## 4  Algorithm

This section describes our algorithm. First, we briefly introduce the *optimal recovery* algorithm from signal processing, which is a fundamental building block of our approach. Then, we describe our estimation strategy.

Optimal recovery was introduced to effectively approximate a function known to belong to a certain signal class with limited information about it (Micchelli & Rivlin, 1976). It has been applied to estimate missing or corrupted pixels in images (Muresan & Parks, 2004) and missing values in biological data (Dean & Varshney, 2021). Optimal recovery estimates the missing value using a learned principal vector and a set of known values. In this framework, the known values, such as sampled pixels or pre-intervention observations, are *linear functionals*. Linear functionals refer to any operation that maps a signal (image patch) to a scalar via linear mappings, e.g. derivatives, averages Muresan & Parks (2001). The representers, derived via the Riesz representation theorem, then map these known linear measurements back into the full signal space, allowing reconstruction imposing the structure of observed data.

One way the principal vector is constructed is using an ellipsoidal signal class of vectors that pass through a hyperplane.

**Definition 1 (Ellipsoids)** *Given a matrix $Q$, an ellipsoid $K$ is a bounded convex set which has the form*

$$K = \{u \in \mathbb{R}^n : u^\top Q u \leq h\} \tag{2}$$

*where $Q = Q^\top$, and $Q \succ 0$. That is, $Q$ is symmetric and positive definite (Boyd & Vandenberghe, 2004).*

The value $h$ is the ellipsoid radius. The matrix $Q$ determines how far the ellipsoid extends in every direction from the center (the ellipsoid semi-axis lengths are determined by eigenvalues of $Q$).

### 4.1 Ellipsoidal Optimal Recovery (EOpR)

Here, we extend *optimal recovery* with ellipsoidal signal class to estimate causal effects in panel data. Under the potential outcomes framework, we treat the estimation of causal effects as a missing data problem (Athey et al., 2021). Therefore, we use optimal recovery from signal processing and approximation theory to recover the missing outcomes for causal inference, a setting in which optimal recovery has not been considered before.

Geometrically, we assume control units vectors in $\boldsymbol{A}$ belong to an ellipsoidal class $K$, and *a fortiori* rather than by assumption, the pre-intervention of control units $\boldsymbol{A}^-$ belong to a hyperplane $\mathcal{H}$. We consider the treated unit $a_1$ as a partially-observed vector, where $a_1^+ = \{a_{1t}\}_{t > T_0}$ (in the post-intervention) is unknown and requires approximation. The vector $a_1$ lies in $C$, the intersection of $K$ and $\mathcal{H}$.

The aim is to find an estimator that minimizes the worst-case error. This is equivalent to finding the Chebyshev center of $C$ (Boyd & Vandenberghe, 2004). Figure 1 illustrates the geometry of optimal recovery.

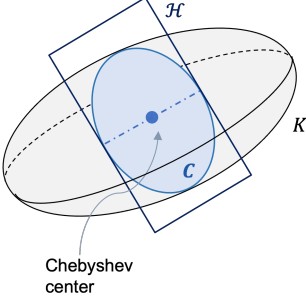

Figure 1: Geometric illustration for the optimal recovery algorithm with hyperplane $\mathcal{H}$ intersecting ellipsoid $K$, creating ellipse $C$, with a Chebyshev center

The benefit of the Chebyshev center is that it provides a minimax optimal solution for the recovery problem (Boyd & Vandenberghe, 2004), as in the following theorem.

**Theorem 1 (Minimax Optimality)** *Let $C \in \mathbb{R}^n$ be an ellipse that represents an intersection of an ellipsoid and a hyperplane. Ellipse $C$ is a bounded and convex set with nonempty interior. A Chebyshev center is a point inside $C$ and is the minimal farthest distance from other points in $C$. For a given point $a_1 \in C$,*

*the estimator $\hat{a}_1 \in C$*

$$\min_{\hat{a}_1} \max_{a_1 \in C} ||\hat{a}_1 - a_1||. \tag{3}$$

*is a Chebyshev center and it is the minimax estimator for $a_1$.*

*Proof.* Follows directly from the definition of the Chebyshev center (Boyd & Vandenberghe, 2004; Micchelli & Rivlin, 1976).

So, our main assumption is that the true signal $a_1$ lies within an ellipsoid shape (intersection of a hyperplane and a higher-dimensional ellipsoid). This acts as a shape constraint on the treated unit, asserting that its underlying structure is similar to that of the controls (within the ellipsoid and hyperplane). This aligns with smoothness and low-complexity assumptions common in matrix completion used in causal inference. Further, having $a_1$ lying in the intersection follows the minimax optimality definition in which we can estimate the counterfactual.

**Connection to PCA**    Constructing the ellipsoid in some way is similar to principal component analysis (PCA). Specifically, the PCs of the data matrix $\boldsymbol{A}$ are in the direction of the most stretch of the ellipsoid $K$. The PC vectors are the eigenvectors corresponding to the largest eigenvalues of $\boldsymbol{A}\boldsymbol{A}^\top$. PCA have been employed in previous work for causal inference (Agarwal et al., 2020a; 2023), to estimate missing outcomes. Though learning the ellipsoid is similar to low-rank estimation, however, we also employ convex inner products (as we see next) to learn a min-max optimal estimator which differs from previous works.

### 4.2    Estimation

To find the Chebyshev center, i.e. minimax estimator $\hat{a}_1$, we take two steps: (1) **Learning** which includes learning the ellipsoidal class $K$ as in equation 2, and learning the *representers*, which we will elaborate on next. (2) **Extrapolation** given the learned spaces and the partially-known vector $a_1^-$.

#### 4.2.1    Learning

**(1) Learning the ellipsoid**    We start by constructing a covariance matrix $\boldsymbol{\Sigma}$ from the data matrix $\boldsymbol{A}$, such that $\boldsymbol{\Sigma} = \boldsymbol{A}\boldsymbol{A}^\top + \lambda \boldsymbol{I}$, where $\boldsymbol{\Sigma} \in \mathbb{R}^{T \times T}$, $\lambda$ is a scalar, and $\boldsymbol{I}$ is the identity matrix. Selection of $\lambda$ is further detailed in Appendix 8.2.

To learn the ellipsoidal class $K$ in equation 2, $K$ must have the most stretch in the same direction of $\boldsymbol{\Sigma}$, hence we let the eigenvalues of $\boldsymbol{Q}$ in equation 2 be the reciprocal of the eigenvalues of $\boldsymbol{\Sigma}$,

$$\boldsymbol{Q} = \boldsymbol{\Sigma}^\dagger. \tag{4}$$

**(2) Learning the representers**    Given the pre-intervention matrix $\boldsymbol{\Sigma}^- \in \mathbb{R}^{T \times T_0}$, we derive the representers $\boldsymbol{\Phi} \in \mathbb{R}^{T_0 \times T_0}$, by the Riesz representation theorem (Muresan & Parks, 2004), as

$$\boldsymbol{\Phi} = \boldsymbol{\Sigma}^{-\top} \boldsymbol{Q} \boldsymbol{\Sigma}^-. \tag{5}$$

We can think of representers, in geometric terms, as the vectors passing through $\boldsymbol{Q}$ in the direction of the ellipsoid. The time samples of the pre-intervention period are, by construction, linear functionals of the full signals. Such that, we observe the first $T_0$ time samples of the pre-intervention period, we consider them as fixed coordinate projections of the full signal vector. These projections define an affine constraint, then the Riesz representation theorem guarantees that there exists a unique set of vectors—the representers—that map these constraints into the signal space via the induced inner product. (Further details on the Riesz representation theorem are in Appendix 8.1). Given that, we can extrpolate as in the next step.

#### 4.2.2    Extrapolation

Now we calculate $\hat{a}_1$, the Chebyshev center of $C$. Given that the representer vectors lie in a subspace that is parallel to the center of the ellipse $C$, and a known vector $a_1^-$, $\hat{w}$ is a linear combination of the inverse of

representers $\boldsymbol{\Phi}$, such that

$$\hat{w} = (\boldsymbol{\Phi})^{-1} a_1^-, \tag{6}$$

then, given the pre-intervention of controls, $\boldsymbol{\Sigma}^-$, and weights $\hat{w}$, the Chebyshev center, the estimated outcome $\hat{a}_1$ is

$$\hat{a}_1 = \boldsymbol{\Sigma}^{-\top} \hat{w}. \tag{7}$$

Geometrically, we consider $\Sigma^- \in R^{T \times T_0}$ to be in the hyperplane H, and K to be the set of representers vectors in $\boldsymbol{\Phi}$ (equation 5). The representers in $K$ yield a feasible set of signals whose projection onto the pre-intervention vectors match the observed pre-intervention values. Therefore, the intersection of $K$ and $\mathcal{H}$, creates an induced subset $C$ that contains full signals that are consistent with the prior covariance structure (of ellipsoid $K$) and match the observed pre-intervention trajectory (by projection onto representers).

**Estimate true vector $s_1$:** Recall the latent factor model in equation 1, where $x_{it} = s_{it} + \epsilon_{it}$. To estimate the vector of interest $s_1$, we apply the learning step on $\boldsymbol{X}$ (in place of the generic $\boldsymbol{A}$), and extrapolate using the actual partially observed values in $s_1^-$, which is the pre-treatment vector. This leads to $\hat{s}_1$, which is an optimal Chebyshev solution. Hence, we rewrite equation 6-7 as the following.

$$w^* = (\boldsymbol{\Phi})^{-1} s_1^-, \tag{8}$$

and

$$\hat{s}_1 = \boldsymbol{\Sigma}^{-\top} w^*. \tag{9}$$

### 4.3 Worst-case estimates

In situations of deep uncertainty, cf. Lempert et al. (2006), it is important to acknowledge the most severe possible outcomes that could occur for a given policy. In deep uncertainty, one is quite concerned about distinguishing what is possible from what is impossible, especially with limited knowledge about future conditions.

EOpR can also estimate the worst-case scenario as follows.

Given that the estimator $\hat{a}_1$ is at the center of the intersection $C$, the vectors on the boundary of $C$ are the worst-case estimates. To attain worst-case vectors (minimax control of the estimates), let $y$ be the unit norm in $\mathcal{Z}$, a parallel subspace to representers $\boldsymbol{\Phi}$, determined by

$$y = \boldsymbol{\Phi}^{-1} \boldsymbol{\Sigma}^{-\top} \boldsymbol{Q} \boldsymbol{\Sigma}. \tag{10}$$

The worst-case estimates $\bar{a}_1$ are:

$$\bar{a}_1 = \hat{a}_1 \pm (\varepsilon - \|\hat{a}_1\|_Q)^{\frac{1}{2}} y, \tag{11}$$

with a very small $\varepsilon$, and the $\boldsymbol{Q}$-norm $\|\hat{a}_1\|_Q = \hat{a}_1^\top Q \hat{a}_1$.

### 4.4 Properties of the estimator

In higher dimensions, overlap and extrapolation might become geometrically challenging. Indeed, balancing weight methods, e.g., Abadie & Gardeazabal (2003); Bruns-Smith & Feller (2022) often fail if the treated unit lies outside the convex hull and interpolation becomes unstable.

In our case, rather than relying on interpolation within the convex hull, we have this feasible region resulting from intersection. While it is true that in some cases the intersection can become small, however, our formulation ensures that the intersection is non-empty, by construction, and the solution is well-defined. Recall, the hyperplane is constructed from full signals of controls, and the ellipsoid is the pre-intervention of controls, therefore, in panel data setting, intersection is non-empty.

Also, unlike the convex hull, given that the ellipsoid expands in direction of high variance, it is not restricted to positive convex combinations, it can extrapolate along principal directions. Furthermore, the presence of $\lambda$ is substantial: the ellipsoid contracts in low-variance direction but never collapses due to the addition of ridge term to the covariance.

**Consistency & Unbiasedness** The extrapolation step produces an estimator that is the Chebyshev center of the ellipse $C$ with desirable properties of unbiasedness and consistency. We measure performance using mean-squared error (MSE).

**Theorem 2** *Let $\tilde{r}$ denote the rank of matrix $\mathbf{\Sigma}^-$, for $\lambda > 0$, and random noise $\epsilon$ with variance $\sigma^2$. Then the estimation error can be bounded as*

$$MSE(s_1, \hat{s_1}) \leq \frac{2\sigma^2 \lambda_{max} \tilde{r}}{T}, \tag{12}$$

*such that $\tilde{r} \leq T_0 < T$, and $\lambda_{\max}$ is the largest eigenvalue of $\mathbf{\Sigma}^-$.*

***Proof.** **1** Proof is in Appendix 9.*

We say that the estimator is *consistent* when the MSE converges to zero if $T$ grows without bound. The value $\tilde{r}$ is strictly less than $T$. An estimator is said to be *unbiased* when the estimated vector equals the true vector on average.

The Chebyshev center has previously been proven to be a consistent and unbiased estimator geometrically (Halteman, 1986) for spherically-symmetric noise distributions $\epsilon$, but the argument extends directly for non-spherical noise, full proof is in Appendix 9.3.

## 5 Empirical Analysis

We compare the accuracy of our EOpR approach with other causal inference methods used in policy evaluation: SC (Abadie & Gardeazabal, 2003), RSC (Amjad et al., 2018), DSC (Ferman & Pinto, 2021), and SDID (Arkhangelsky et al., 2019). We first evaluate on simulated data to demonstrate properties of EOpR under certain settings. We also consider empirical simulations of the Penn World Table simulations based on Arkhangelsky et al. (2019) in Appendix 10.2.

We then evaluate on two classical panel datasets commonly used in the SC literature (California Proposition 99 (Abadie et al., 2010) and Basque Country (Abadie & Gardeazabal, 2003)). We finally apply EOpR in the context of the COVID-19 pandemic to estimate the number of confirmed cases in New York State.

### 5.1 Evaluation Metrics

To measure the quality of estimation, we use two metrics. First, we measure the root-mean-square error (RMSE) of estimated vectors. The pre-intervention (training) error is for $1 \leq t \leq T_0$, and a post-intervention (testing) error is for $T_0 < t \leq T$:

$$\text{RMSE}(u, \hat{u}) = \left( \frac{1}{\mathcal{T}} \sum_{t=1}^{\mathcal{T}} (u - \hat{u})^2 \right)^{1/2}, \tag{13}$$

where $\mathcal{T}$ is the size of the selected time period, and $u$ and $\hat{u}$ are two dummy variables.

Second, Abadie et al. (2010) proposed a test statistic to evaluate the reliability of the estimates by running *placebo tests*. One placebo test considers one control unit as a placebo treated unit and apply the estimation algorithm. Since control units are assumed to not be affected by the examined intervention, one would expect that the estimated signal for the placebo unit does not diverge from its corresponding control unit. Further, the gaps between each placebo estimation and its corresponding control unit should be less divergent than the gap between the original treated unit and its estimation. Placebo tests are applied on the classical econometrics case studies.

### 5.2 Simulations

We conduct synthetic simulations to demonstrate the properties of EOpR estimates in both the pre- and post-intervention periods. We show that EOpR performs well and better than existing causal inference methods under various settings.

**Experimental setup.** Consider a data generating process, which is frequently considered for low-rank matrix decomposition solutions, similar to Amjad et al. (2019), as follows. First we create two sets of row and column features, $B_r$, $B_c$, where $B_r = \{b_k | b_k \sim \text{Unif}(0,1), 1 \leq k \leq 10\}$ and $B_c = \{b_k | b_k \sim \text{Unif}(0,1), 1 \leq k \leq 10\}$. For each unit $2 \leq i \leq N$, we assign a parameter $\theta_i$ drawn from $B_r$ (with replacement), and for each time $1 \leq t \leq T$ we assign a parameter $\rho_t$ drawn from $B_c$ (with replacement). We use the following formula to generate a data point $\tilde{s}_{it} = f(\theta_i, \rho_t)$ to construct the control units

$$f(\theta_i, \rho_t) = \frac{10}{1 + \exp\left(-\theta_i - \rho_t - (\theta_i \rho_t)\right)} + \epsilon_{it}, \tag{14}$$

where $\epsilon_{it} \sim \mathcal{N}(0,1)$, an independent Gaussian noise.

The data-generating process follows the latent factor model in equation 1, such that we are generating noisy $X$ variables and applying the algorithm to them, but measuring performance for the true mean vector $s_1$.

In the following experiments, we investigate EOpR resistance to bias in comparison to other algorithms under different numbers of units $N$ and time periods $T_0$ and $T$. For each combination of $N$, $T_0$, and $T$ we generate ten simulations and average the resulting RMSE scores for the estimated pre- and post-intervention signals of $\tilde{s}_{1t}$.

### 5.2.1 Length of pre-intervention period

When the time of intervention $T_0$ starts very early in a period of time, it creates a short pre-intervention period. It has been discussed earlier that if $T_0$ is too short it fails to reproduce the trajectory of the treated unit in synthetic control methods (Abadie, 2021). Here, we test the robustness of our method under very limited length of pre-intervention period $T_0$. We fix the size of units $N$ and post-intervention length $T_1$. We vary the time of intervention $T_0$ to be as short as 10% or as large as close to $T_1$.

Figure 2 shows the effect of different pre-treatment lengths on the algorithm's ability to estimate, fixing $N = 50$ and $T_1 = 50$. When having either a small or large number of pre-treatment periods, EOpR recovers the original treated signal with the smallest error compared to other algorithms. Specifically at $T_0 \leq 30$, where the pre-treatment period is relatively short, EOpR extrapolates beyond the training periods with the least estimation bias, whereas other algorithms have more bias in the post-intervention estimation.

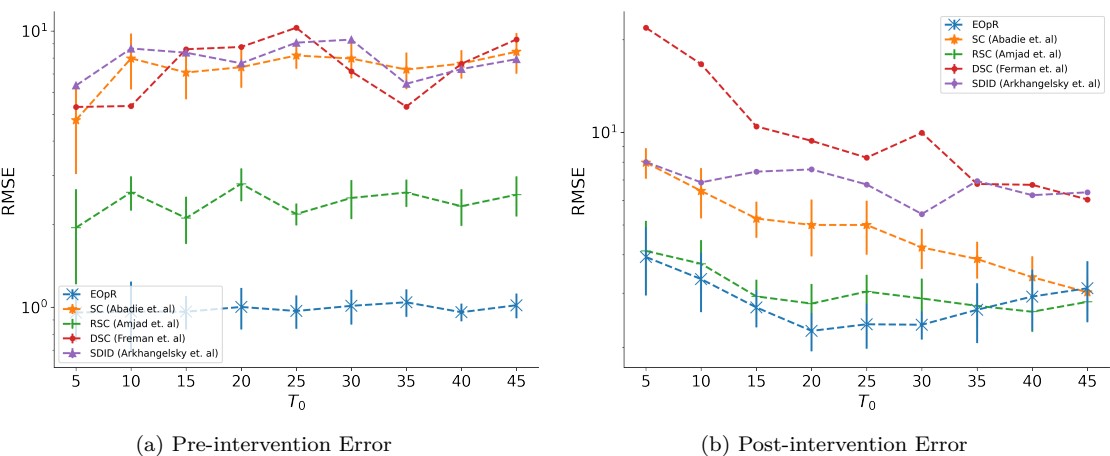

(a) Pre-intervention Error        (b) Post-intervention Error

Figure 2: Growing size of pre-intervention period $T_0$ with fixed post-intervention $T_1 = 50$ and $N = 50$

### 5.2.2 Length of post-intervention period

We further investigate the ability of EOpR to estimate the trajectory of the post-intervention for an extended period of time. The ability to estimate for an extended period of time indicates an algorithm is robust and consistent as $T \rightarrow \infty$ .

Figure 3 shows the estimation errors with fixed $N = 100$ and $T_0 = 50$, and tested over multiple lengths of post-intervention. EOpR consistently achieves a low estimation error in both pre- and post-intervention estimation, especially at longer periods of time, e.g. ($t = 450$).

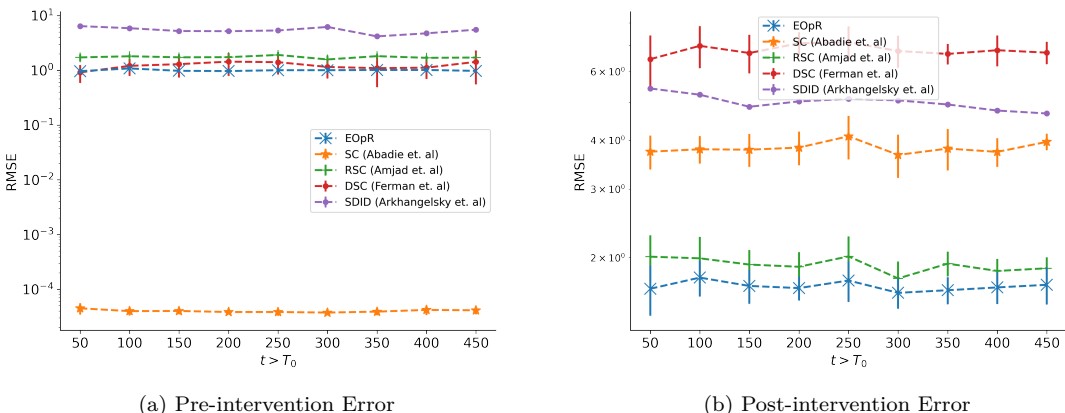

(a) Pre-intervention Error            (b) Post-intervention Error

Figure 3: Growing post-intervention periods, with $N = 100$ and $T_0 = 50$

Further simulation experiments are in Appendix 10.1. We also consider simulations with an added non-smoothness structure to test the robustness of our method in Appendix 10.1.3, where EOpR outperformed other baselines in most non-smooth settings.

### 5.3 Real-World Experiments

We explore two econometric real-world case studies: the Basque Country case study Abadie et al. (2010) and the California 99 Proposition case study Abadie & Gardeazabal (2003) (in Appendix 10.3). Both case studies showcase the ability of the original synthetic control estimator to produce reliable counterfactual estimation. We use the two case studies to demonstrate the robustness of our proposed algorithm in producing a reliable counterfactual. Given that the ground truth is missing in these case studies, econometricians consider placebo tests to show reliability of estimator, and we follow the same procedure for evaluation. We also consider a third case-study that we develop for COVID-19 data and lockdown interventions.

*(Note that our method returns worst-case estimates, whereas baseline methods consider statistical notions of uncertainty. Therefore they are not directly comparable)*

#### 5.3.1 Basque Country

The objective of this case study is to investigate the effect of terrorist attacks on the Basque Country economy compared to other Spanish regions. Terrorist activities started by 1970. There was a significant negative impact on the economy of Basque Country measured by per-capita GDP. Abadie et al. (2010) showed that the economy would be better without terrorism.

**Results.** Figure 4 shows the actual trajectory of the Basque Country economy in black, with a degradation after 1970. In comparison to other methods, our EOpR method recovers Basque Country estimate, with more accurate fit on the pre-intervention values of the treated unit. Figure 4 shows the estimated worst-case potential outcomes from EOpR. Given the noise inherited in historical data and the extrapolation beyond observed inputs, worst-case estimates may look loose, however, the difference between mean estimate and bounds is not abnormally large, about 20-25 units, which is within an order of magnitude, and reflects a reasonable level of uncertainty.

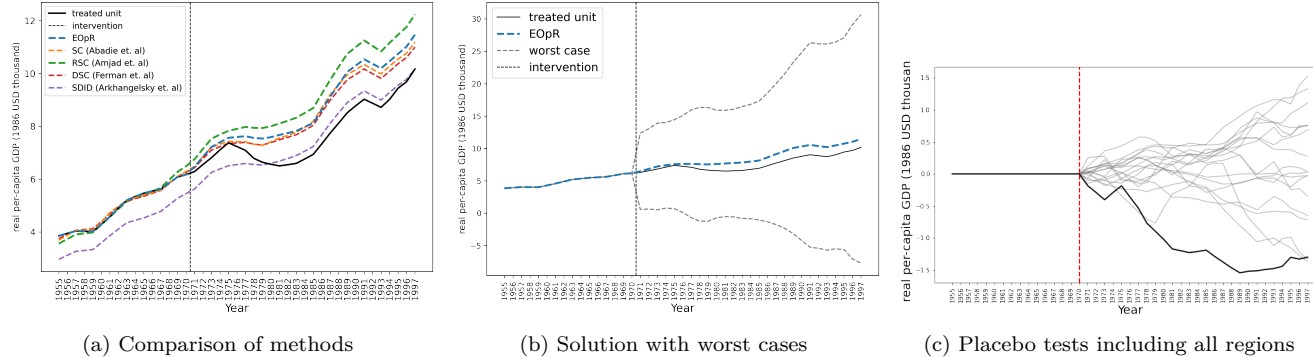

(a) Comparison of methods     (b) Solution with worst cases     (c) Placebo tests including all regions

Figure 4: Trends in per-capita GDP for Basque Country and synthetic Basque Country and Placebo tests

**Placebo Tests.** We create placebo tests, similar to Abadie et al. (2010). Note that Abadie et al. (2010) excluded five regions that had poor fit in the pre-intervention, but we keep all regions. We plot the differences between our estimates and the observations of all regions as placebo and Basque Country (the actual treated unit). Figure 4(c) shows the differences for all regions compared to Basque Country (solid black line in the figure). The divergence for Basque Country was the largest, thus, the derived estimates by the EOpR are reliable.

### 5.3.2 COVID-19 in New York

New York was one of the earliest American states that turned to an epicenter of COVID-19 during 2020 (Thompson et al., 2020). We aim to estimate the New York COVID-19 cases trajectory with states as control units using EOpR and other methods. We select the states where lockdowns were imposed, yielding a total of 43 states.

**Experimental setup** For lockdown dates, we use data from the COVIDVis project[2] that tracks policy interventions at the state level. We consider the dates of *shelter-in-place* mandates. For COVID-19 cases, we consider state-level case load data from The New York Times (2021). Note that since reported cases depend on COVID-19 testing, our analysis is limited by the fact there was widespread shortage of available tests in different regions at different times.

Since each state imposed a lockdown at different times, we aligned states based on the days differences between the time a lockdown took place and the rest of the dates during the period of interest, following Bayat et al. (2020). Figure 5 shows the trend of New York and seven other states and their moving averages (over 7 days).

Based on the SC literature (Abadie et al., 2010), one would need to select the control units based on their similar trends to the target unit to ensure a correct construction of the treated unit. In the following experiments, we estimate New York COVID-19 cases using 43 states, for a period of 200 days, $T = 200$ to approximate the period between March and August in 2020. We vary the lengths of pre-treatment period by varying the time of intervention $T_0 \in \{5, 10, 15, 20, 25, 30\}$ days.

In this experiment, we estimate the "factual" trend instead of the "counterfactual", given that we already have the factual trend of New York as the post-intervention, which is the ground truth. Given the ground truth and the estimated trend by algorithms, we compute the error. This method of testing the reliability of estimation algorithms has been previously used in various works (Bayat et al., 2020).

**Results.** Table 1 shows a comparison of estimation performance of our algorithm and the other four algorithms. At short pre-intervention periods, e.g. $T_0 = 5$ and $T_0 = 10$, EOpR has produced the lowest RMSE in pre- and post-treatment trends in comparison to other algorithms. This reflects its ability to

---

[2]https://covidvis.berkeley.edu/#lockdown_section

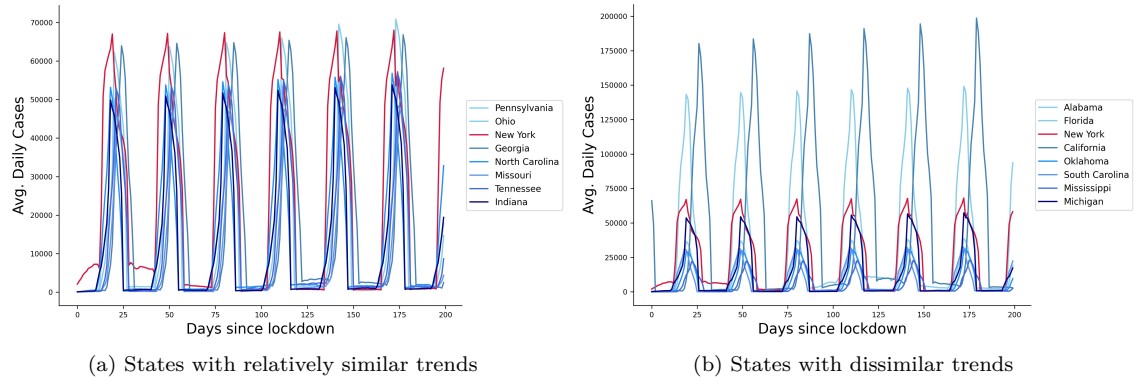

(a) States with relatively similar trends     (b) States with dissimilar trends

Figure 5: Moving average of COVID-19 confirmed cases of New York (in red) and 7 other states

Table 1: Pre- and Post-intervention RMSE for COVID-19 case study

| | Pre-intervention (Training) Error | | | | | |
|---|---|---|---|---|---|---|
| $T_0$ | 5 | 10 | 15 | 20 | 25 | 30 |
| EOpR | **0.017** | **0.021** | 0.631 | **0.499** | 0.706 | 0.831 |
| SC (Abadie et al.) | 0.043 | 0.074 | **0.316** | 0.501 | **0.501** | **0.525** |
| RSC (Amjad et al.) | 0.094 | 0.139 | 0.434 | 0.905 | 0.890 | 0.834 |
| DSC (Ferman et al.) | 0.061 | 0.129 | 0.377 | 0.745 | 0.659 | 0.685 |
| SDID (Arkhangelsky et. al) | 0.045 | 0.063 | 0.397 | 0.922 | 1.005 | 0.964 |
| | Post-intervention (Testing) Error | | | | | |
| $T_0$ | 5 | 10 | 15 | 20 | 25 | 30 |
| EOpR | **0.842** | **0.852** | 0.676 | **0.456** | 0.633 | 0.803 |
| SC (Abadie et al.) | 0.921 | 0.959 | 0.896 | 0.512 | **0.491** | **0.477** |
| RSC (Amjad et al.) | 0.874 | 0.893 | 0.936 | 0.854 | 0.788 | 0.790 |
| DSC (Ferman et al.) | 0.958 | 0.879 | **0.602** | 0.730 | 0.611 | 0.645 |
| SDID (Arkhangelsky et. al) | 0.947 | 0.958 | 0.964 | 0.935 | 0.922 | 0.926 |

recover the true estimates even under limited time in pre-intervention, or for an early intervention decision. Figure 15 (in appendix) shows the estimation trajectories for each choice of $T_0$.

## 6 Discussion

Theoretically and empirically, we demonstrate desirable properties of the EOpR algorithm in settings of counterfactual estimation. EOpR differs from SC and its variants in three main ways.

First, synthetic control (SC) estimates a counterfactual treated unit by linearly interpolating on control units. Contrarily, EOpR estimates a counterfactual treated unit by extrapolating on the covariance of control units. Having the covariance instead of raw control units captures inter-correlation between data points, which makes it more robust for estimation in noisy settings.

Second, SC constrains the weights to sum to unity so they lie on the probability simplex; other SC variants such as DSC relax this assumption with unconstrained weights, whereas RSC allows weights to be negative using ($L_1$ or $L_2$) regularization. On the other hand, EOpR uses the Riesz representation theorem to construct the weights with an inner product of covariance matrices. In other words, the weights are a linear combination of representors, which are obtained by the Riesz representation theorem. The representors are used to represent inner products of covariance matrix.

Third, stemming from the Riesz representation theorem, EOpR derives worst-case estimates, which are two vectors on the farthest edges of the ellipsoid, in the direction of the center. The worst-case estimates can be thought of as bounding the estimation of the treated unit. There is no such worst-case characterization that comes from traditional SC or its variants.

Other than standard SC, our approach is related in spirit to recent works that consider minimax optimization by balancing weights. For example, Bruns-Smith & Feller (2022) leverage balancing weights and convex duality to derive causal estimators that minimize the worst-case bias under outcome model assumptions. Similarly, Kallus (2020) introduce the generalized optimal matching (GOM) framework, which considers a broad class of balancing and matching methods to minimize worst-case loss over hypothesis spaces, e.g., reproducing kernel Hilbert spaces (RKHS), while controlling variance through regularization.

In contrast to these balancing-weights-based methods, our method takes a geometric approach where the solution lies within a covariance-induced ellipsoid. Our formulation avoids dependence on explicit outcome function classes or kernel choices, and rather than constructing balancing weights explicitly, we recover the treated unit as the Chebyshev center of the feasible region. As a result, our approach is particularly suited to settings with limited pre-treatment data, limited overlap, or ill-posed weight construction. This offers a complementary perspective to balancing-weight and function-space methods.

EOpR shows outstanding performance under various settings, especially under limited data availability, short pre-treatment periods (See Table 1) or noisy settings (See Table 2). However, it fails under certain settings. For example, if the treated unit deviates largely from the controls trends, in which it is structurally divergent from the ellipsoid, then the ellipsoid might not contain the true trajectory (shown in the COVID-19 experiment when increased $T_0$ when treated unit starts to deviate from other controls). Also, EOpR fails when pre-intervention controls have a square-shape matrix, such as $N = T_0$. In this case, the matrix is square and invertible which would create an exact interpolation of the pre-treatment treated unit. However, it creates an unstable feasible set, such that the Chebyshev center becomes sensitive to noise.

Additionally, the current version of EOpR does not explicitly account for unobserved confounding factors that may influence both the treatment assignment and the outcomes, such that EOpR does not account for biases introduced by confounders. Recent work in the synthetic control literature such as Agarwal et al. (2020b); Imbens & Viviano (2023) explore ways to incorporate structural assumptions to account for unobserved confounding. Extending our approach to handle confounders is a promising direction for future work.

## 7 Conclusion

Classical synthetic control has been noted as effective for causal inference in comparative studies. Here, we propose an approximation-theoretic approach for synthetic control—ellipsoidal optimal recovery (EOpR)—that estimates the causal effect given a policy intervention. Furthermore, given the properties of EOpR, we derive worst-case estimates, which are themselves very useful for policy evaluation, especially in settings of deep uncertainty. A limitation in EOpR is that given the ellipsoid assumption (a common assumption in econometrics (Armstrong & Kolesár, 2018)), it is important to correctly specify the shape of the ellipsoid, which we do from data. Real-world data may exhibit heteroskedasticity or other non-quadratic characteristics, in which data cannot be modeled adequately using an ellipsoid (but optimal recovery can be extended to other geometric priors (Dean & Varshney, 2021)). Our approach of EOpR has less estimation error for pre- and post-intervention periods, especially with short pre-intervention periods. This is demonstrated through comparisons on simulated data, classical case studies in econometrics, and a new health-relevant setting.

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

# 8 Appendix

## 8.1 Riesz Representation Theorem

The Riesz representation theorem is fundamental in functional analysis, it represents continuous linear functionals as inner products with specific vectors in the space. It establishes a deep connection between Hilbert spaces and their duals. Let $H$ be a Hilbert space over $Z$ and a function $\delta : H \to Z$ is called a linear functional.

**Theorem 3** *(Riesz representation theorem). For a continuous linear functional $\delta$ on a Hilbert space $H$, there exists a unique $h \in H$ such that $\delta(x) = \langle x, h \rangle$, $x_i$ for all $x \in H$. Furthermore, $\|u\| = \|\delta\|$.*

Further details are given by Adler (2021).

## 8.2 Hyperparameter Tuning

Based on the ellipsoid definition in equation 1, the matrix $Q$ must be positive definite. To ensure that eigenvalues of $Q$ are strictly positive, we add a small perturbation $\lambda \in (0, 1]$ to the diagonal of $AA^\top$. We use cross-validation on the pre-intervention period to select $\lambda$ that minimizes the $\ell_2$-norm. Therefore, such covariance matrix $AA^\top$ is used to construct $Q$ and the ellipsoid based on the definition in equation 1.

# 9 Consistency and Unbiasedness

## 9.1 Analytical Proof (Unregularized Covariance)

To show consistency, here we bound the $\ell_2$ error of the estimation. We will drop the dependency on $\lambda$ for the sake of simplicity in proof. Recall that the noise term $\epsilon_1$ is a zero-mean independent random variable that satisfies $\mathbb{E}(\epsilon_{ij}) = 0$ for all $i$ and $j$ by assumption, with variance $Var(\epsilon_{ij}) = \sigma^2$.

**Lemma 1** *Suppose $x_1 = s_1 + \epsilon_1$ with $\mathbb{E}(\epsilon_{1j}) = 0$ and $Var(\epsilon_{1j}) \leq \sigma^2$ for all $j \in \{1, \ldots, T\}$. Let $w^*$ be the min-max optimizer for $\hat{s}_1$. Then for any $\lambda \geq 0$,*

$$\mathbb{E}\|s_1 - \hat{s}_1\| \leq 2\sigma^2 r. \tag{15}$$

*Proof.* (Unregularized Covariance) Recall from equation 1 that the treatment row is $x_1 = s_1 + \epsilon_1$. By definition of the Chebyshev center and its properties of unique estimation, consider $\hat{s}_1 = \Sigma_\phi^\top w^*$ from equation 9. Here, $\Sigma_\phi = AA^\top$. By the generic definition in equation 7, $\hat{w} = \Sigma^{\dagger^\top} x_1$, ($x_1$ in place of the generic $a_1$), and so, $\hat{w}$ is sub-optimal for $\hat{s}_1$.

$$
\begin{aligned}
\|s_1 - \hat{s}_1\|^2 &= \|(x_1 - \epsilon_1) - \Sigma_\phi^\top w^*\|^2 \\
&\leq \|(x_1 - \Sigma_\phi^\top \hat{w} + (-\epsilon_1)\|^2 \\
&\leq \|x_1 - \Sigma_\phi^\top \hat{w}\|^2 + \|\epsilon_1\|^2 + 2\langle -\epsilon_1, x_1 - \Sigma_\phi^\top \hat{w} \rangle \\
&\leq \|(\Sigma_\phi^\top \hat{w} + \epsilon_1) - \Sigma_\phi^\top \hat{w}\|^2 + \|\epsilon_1\|^2 + 2\langle -\epsilon_1, x_1 - \Sigma_\phi^\top \hat{w} \rangle \\
&\leq 2\|\epsilon_1\|^2 + 2\langle -\epsilon_1, x_1 - \Sigma_\phi^\top \hat{w} \rangle.
\end{aligned}
\tag{16}
$$

Taking expectations, we arrive at

$$\mathbb{E}\|s_1 - \hat{s}_1\|^2 = 2\mathbb{E}\|\epsilon_1\|^2 + 2\mathbb{E}\langle -\epsilon_1, x_1 - \Sigma_\phi^\top \hat{w} \rangle. \tag{17}$$

With the noise having variance $\sigma^2$, then

$$\mathbb{E}\|\epsilon_1\|^2 = \sigma^2. \tag{18}$$

We must address an inner product on the right side. First, we derive some useful facts. Recall the trace operator has the mapping property $tr(AB) = tr(BA)$, and the projection matrix $P$ to be $P_1 = AA^\dagger$ and $P_2 = A^\dagger A$. Hence,

$$
\begin{aligned}
\mathbb{E}[(\epsilon_1)^\top \mathbf{\Sigma}_\phi^\top \mathbf{\Sigma}_\phi^{\dagger^\top} \epsilon_1] &= \mathbb{E}[tr((\epsilon_1)^\top \mathbf{\Sigma}_\phi^\top \mathbf{\Sigma}_\phi^\dagger \epsilon_1)] \\
&= \mathbb{E}[tr(\mathbf{\Sigma}_\phi^\top \mathbf{\Sigma}_\phi^\dagger)\epsilon_1(\epsilon_1)^\top] \\
&= tr(\mathbb{E}[\mathbf{\Sigma}_\phi^\top \mathbf{\Sigma}_\phi^{\dagger^\top} \epsilon_1(\epsilon_1)^\top]) \\
&= tr(\mathbb{E}[\mathbf{\Sigma}_\phi^\top \mathbf{\Sigma}_\phi^{\dagger^\top}]\mathbb{E}[\epsilon_1(\epsilon_1)^\top]) \\
&= tr(\mathbb{E}[\mathbf{\Sigma}_\phi^\top \mathbf{\Sigma}_\phi^{\dagger^\top}]\mathbf{\Sigma}_\phi^2 I) \\
&= \sigma^2 \mathbb{E}[tr(\mathbf{\Sigma}_\phi^\top \mathbf{\Sigma}_\phi^{\dagger^\top})] \\
&= \sigma^2 \mathbb{E}[\text{rank}(\mathbf{\Sigma}_\phi)] \\
&\leq \sigma^2 r,
\end{aligned}
\tag{19}
$$

which follows since the trace of a projection matrix equals the rank of the matrix, i.e. $tr(\mathbf{\Sigma}_\phi^\top \mathbf{\Sigma}_\phi^{\dagger^\top}) = \text{rank}(\mathbf{\Sigma}_\phi^\top) = \text{rank}(\mathbf{AA}^\top) = r$. Hence, the rank of $\mathbf{\Sigma}_\phi$ is at most $r$.

Returning to the inner product, by the generic definition in equation 7, $\hat{w} = \mathbf{\Sigma}_\phi^{\dagger^\top} x_1$, ($x_1$ in place of the generic $a_1$). Recall from equation 1, $x_1 = s_1 + \epsilon_1$, then

$$
\begin{aligned}
\mathbb{E}[\langle -\epsilon_1, x_1 - \mathbf{\Sigma}_\phi \hat{w}\rangle] &= \mathbb{E}[(\epsilon_1)^\top \mathbf{\Sigma}_\phi \hat{w}] - \mathbb{E}[(\epsilon_1)^\top x_1] \\
&= \mathbb{E}[(\epsilon_1)^\top \mathbf{\Sigma}_\phi^\top \mathbf{\Sigma}_\phi^{\dagger^\top} x_1] - \mathbb{E}[(\epsilon_1)]s_1 - \mathbb{E}[(\epsilon_1)^\top \epsilon_1] \\
&= \mathbb{E}[(\epsilon_1)^\top \mathbf{\Sigma}_\phi^\top \mathbf{\Sigma}_\phi^{\dagger^\top}]s_1 + \mathbb{E}[(\epsilon_1)^\top \mathbf{\Sigma}_\phi^\top \mathbf{\Sigma}_\phi^{\dagger^\top} \epsilon_1] - \mathbb{E}[(\epsilon_1)^\top \epsilon_1] \\
&= \mathbb{E}[(\epsilon)^\top][\mathbf{\Sigma}_\phi^\top \mathbf{\Sigma}_\phi^{\dagger^\top}]s_1 + \mathbb{E}[(\epsilon_1)^\top \mathbf{\Sigma}_\phi^\top \mathbf{\Sigma}_\phi^{\dagger^\top} \epsilon_1] - \mathbb{E}[(\epsilon_1)^\top \epsilon_1] \\
&= \mathbb{E}[(\epsilon_1)^\top \mathbf{\Sigma}_\phi^\top \mathbf{\Sigma}_\phi^{\dagger^\top} \epsilon_1] - \mathbb{E}\|\epsilon_1\|^2 \\
&\leq \sigma^2 r - \mathbb{E}\|\epsilon_1\|^2.
\end{aligned}
\tag{20}
$$

Finally, we replace the above terms into inequality equation 17 to arrive at

$$
\begin{aligned}
\mathbb{E}\|s_1 - \hat{s}_1\|^2 &\leq 2\mathbb{E}\|\epsilon_1\|^2 + 2\mathbb{E}\langle -\epsilon, x_1 - \mathbf{\Sigma}_\phi^\top \hat{w}\rangle \\
&\leq 2\mathbb{E}\|\epsilon_1\|^2 + 2\sigma^2 r - 2\mathbb{E}\|\epsilon_1\|^2 \\
&\leq 2\sigma^2 r,
\end{aligned}
\tag{21}
$$

which completes the proof.

Now, let us consider replacing the above value in the mean squared error ($MSE$), which can be bounded by

$$
\begin{aligned}
MSE(s_1, \hat{s}_1) &= \frac{1}{T}\|s_1 - \hat{s}_1\|^2 \\
&\leq \frac{2\sigma^2 r}{T}.
\end{aligned}
\tag{22}
$$

## 9.2 Analytical Proof (Regularized Covariance)

Recall from equation 1 that the treatment row is $x_1 = s_1 + \epsilon_1$. By definition of the Chebyshev center and its properties of unique estimation, consider $\hat{s}_1 = \mathbf{\Sigma}^\top w^*$ from equation 9. Here, $\mathbf{\Sigma} = \mathbf{AA}^\top + \lambda \mathbf{I}$.

By the generic definition in equation 7, $\hat{w} = \Sigma^{\dagger\top} x_1$, ($x_1$ in place of the generic $a_1$), and so, $\hat{w}$ is sub-optimal for $\hat{s}_1$.

$$
\begin{aligned}
\|s_1 - \hat{s}_1\|^2 &= \|(x_1 - \epsilon_1) - \boldsymbol{\Sigma}^\top w^*\|^2 \\
&\leq \|(x_1 - \boldsymbol{\Sigma}^\top \hat{w} + (-\epsilon_1)\|^2 \\
&\leq \|x_1 - \boldsymbol{\Sigma}^\top \hat{w}\|^2 + \|\epsilon_1\|^2 + 2\langle -\epsilon_1, x_1 - \boldsymbol{\Sigma}^\top \hat{w}\rangle \\
&\leq \|(\boldsymbol{\Sigma}^\top \hat{w} + \epsilon_1) - \boldsymbol{\Sigma}^\top \hat{w}\|^2 + \|\epsilon_1\|^2 \\
&\quad + 2\langle -\epsilon_1, x_1 - \boldsymbol{\Sigma}^\top \hat{w}\rangle \\
&\leq 2\|\epsilon_1\|^2 + 2\langle -\epsilon_1, x_1 - \boldsymbol{\Sigma}^\top \hat{w}\rangle.
\end{aligned}
\tag{23}
$$

Taking expectations, we arrive at

$$
\mathbb{E}\|s_1 - \hat{s}_1\|^2 = 2\mathbb{E}\|\epsilon_1\|^2 + 2\mathbb{E}\langle -\epsilon_1, x_1 - \boldsymbol{\Sigma}^\top \hat{w}\rangle.
\tag{24}
$$

The zero-mean noise $\epsilon$ has $\mathbb{E}(\epsilon) = 0$. Also note that $\mathbb{E}(z) = 0$ for any variable $z$ independent of noise. The random noise has variance $\sigma^2$, such that

$$
\mathbb{E}\|\epsilon_1\|^2 = \sigma^2.
\tag{25}
$$

Now, we want to solve for the inner product $2\langle -\epsilon_1, x_1 - \boldsymbol{\Sigma}^\top \hat{w}\rangle$. Recall from equation 5 that $\boldsymbol{\Sigma}^- = \boldsymbol{\Sigma}[:, T_0]$, a pre-intervention column-wise subset of $\boldsymbol{\Sigma}$. For the sake of this proof only, we fix $\boldsymbol{\Sigma}_u = \boldsymbol{\Sigma}^-$. Using equations 4, 5, and 6, we rewrite $\hat{w} = (\boldsymbol{\Sigma}_u^\top \boldsymbol{\Sigma}^\dagger \boldsymbol{\Sigma}_u)^{-1} x_1^-$, replacing generic $a_1^-$ by $x_1^-$.

Let $R \in \mathbb{R}^{T_0 \times T_0}$ as:

$$
R := \boldsymbol{\Sigma}_u^\top \boldsymbol{\Sigma}^\dagger \boldsymbol{\Sigma}_u.
\tag{26}
$$

Consider a matrix $\boldsymbol{P} \in \mathbb{R}^{T_0 \times T}$ that selects the pre-intervention vectors, such that $\epsilon_1^- = \boldsymbol{P}\epsilon_1$.

Going back to the inner product, and using the definition of $\hat{w} = (\boldsymbol{\Sigma}_u^\top \boldsymbol{\Sigma}^\dagger \boldsymbol{\Sigma}_u)^{-1} x_1^-$, and $x_1^- = s_1^- + \epsilon_1^-$, recall $\epsilon_1^- \in \mathbb{R}^{T_0 \times 1}$.

$$
\begin{aligned}
\mathbb{E}\langle -\epsilon_1, x_1 - \boldsymbol{\Sigma}^\top \hat{w}\rangle &= \mathbb{E}[(\epsilon_1)^\top \boldsymbol{\Sigma}\hat{w}] - \mathbb{E}[(\epsilon_1)^\top x_1] \\
&= \mathbb{E}[(\epsilon_1)^\top \boldsymbol{\Sigma}(\boldsymbol{\Sigma}_u^\top \boldsymbol{\Sigma}^\dagger \boldsymbol{\Sigma}_u)^{-1} x_1^-] - \mathbb{E}[(\epsilon_1)]s_1 - \mathbb{E}[(\epsilon_1)^\top \epsilon_1] \\
&= \mathbb{E}[(\epsilon_1)^\top \boldsymbol{\Sigma}(\boldsymbol{\Sigma}_u^\top \boldsymbol{\Sigma}^\dagger \boldsymbol{\Sigma}_u)^{-1} s_1^-] + \mathbb{E}[(\epsilon_1)^\top \boldsymbol{\Sigma}(\boldsymbol{\Sigma}_u^\top \boldsymbol{\Sigma}^\dagger \boldsymbol{\Sigma}_u)^{-1} \epsilon_1^-] - \mathbb{E}[(\epsilon_1)^\top \epsilon_1] \\
&\overset{(a)}{=} \mathbb{E}[(\epsilon_1)^\top \boldsymbol{\Sigma}\boldsymbol{P}^\top R^{-1} \boldsymbol{P}\epsilon_1] - \mathbb{E}[(\epsilon_1)^\top \epsilon_1] \\
&\overset{(b)}{=} \mathrm{tr}(\mathbb{E}[(\epsilon_1)^\top \boldsymbol{\Sigma}\boldsymbol{P}^\top R^{-1} \boldsymbol{P}\epsilon_1]) - \mathbb{E}[(\epsilon_1)^\top \epsilon_1] \\
&= \mathbb{E}[(\epsilon_1)^\top \epsilon_1]\mathrm{tr}(\mathbb{E}[\boldsymbol{\Sigma}\boldsymbol{P}^\top R^{-1} \boldsymbol{P}]) - \mathbb{E}[(\epsilon_1)^\top \epsilon_1] \\
&\overset{(c)}{\leq} \sigma^2 \lambda_{\max} \cdot \mathbb{E}[\mathrm{rank}(R^{-1})] - \mathbb{E}\|\epsilon_1\|^2 \\
&\leq \sigma^2 \lambda_{\max} \tilde{r} - \mathbb{E}\|\epsilon_1\|^2 \\
&\leq \sigma^2 \lambda_{\max} \tilde{r} - \mathbb{E}\|\epsilon_1\|^2 .
\end{aligned}
\tag{27}
$$

In (a) we replace $(\epsilon_1)^\top \boldsymbol{\Sigma}(\boldsymbol{\Sigma}_u^\top \boldsymbol{\Sigma}^\dagger \boldsymbol{\Sigma}_u)^{-1} \epsilon_1^-$ with $(\epsilon_1)^\top \boldsymbol{\Sigma}\boldsymbol{P}^\top R^{-1} \boldsymbol{P}\epsilon_1$ from equation 26. In (b) we recognize that $(\epsilon_1)^\top \boldsymbol{\Sigma}\boldsymbol{P}^\top R^{-1} \boldsymbol{P}\epsilon_1$ is a scalar random variable, which allows us to replace the random variable by its own trace Amjad et al. (2018). Next, in (c) we upper bound the trace with the maximum eigenvalue $\lambda_{\max}$ and the rank of $R$. The $\mathrm{rank}(R^{-1}) = \tilde{r} \leq T_0 < T$.

We combine the expressions as follows:

$$
\begin{aligned}
\mathbb{E}\|s_1 - \hat{s}_1\|^2 &\leq 2\mathbb{E}\|\epsilon_1\|^2 + 2\sigma^2 \lambda_{\max} \tilde{r} - 2\mathbb{E}\|\epsilon_1\|^2 \\
&\leq 2\sigma^2 \lambda_{\max} \tilde{r}
\end{aligned}
\tag{28}
$$

Now, let us consider replacing the above value in the mean squared error (*MSE*), which can be bounded by

$$
\begin{aligned}
MSE(s_1, \hat{s}_1) &= \frac{1}{T}\|s_1 - \hat{s}_1\|^2 \\
&\leq \frac{2\sigma^2 \lambda_{\max} \tilde{r}}{T},
\end{aligned}
\tag{29}
$$

such that $\tilde{r} < T$.

### 9.3 Geometric Proof

The following theorem of Halteman (1986) demonstrates that the Chebyshev center is an unbiased and consistent estimator for the center of a any spherically-symmetric distribution over a $k$-sphere. The result can further be extended from a sphere to an ellipsoid.

**Theorem 4 (Unbiasedness and consistency)** *Suppose the points $W = \{w_i, \ldots, w_n\}$ are sampled from an independent and identically distributed (i.i.d.) spherical k-dimensional distribution to create a sphere $S(\omega, \pi)$, with center $\omega$ and radius $\pi$. Let $S(\bar{\omega}, \bar{\pi})$, be the smallest sphere containing the samples of $W$, (i.e. $\bar{\pi} < \pi$). The center $\bar{\omega}$, a Chebyshev center, has a spherical distribution and hence $\bar{\omega}$ is unbiased for $\omega$, and consistent, i.e. $\bar{\omega} \to \omega$ with probability goes to unity.*

**Proof. 2** *Given that distribution of $S(\omega, \pi)$ is i.i.d., it is invariant under any rotation about $\omega$. The center $\bar{\omega}$, for any rotated sample in $W$, will be rotated by an equal amount of $\omega$. Therefore, the distribution of $\bar{\omega}$ is rotationally invariant about the same $\omega$, and thus the Chebyshev center $\bar{\omega}$ is unbiased.*

*To prove consistency, let the sphere $S(\omega, \pi)$ have density function $f(w) = \frac{1}{n}$, and let $H$ be the set of points at the boundary of $S(\bar{\omega}, \bar{r})$. Then, the maximum distance $d$ between $w \in H$ and the sphere $S(\omega, \pi)$ converges to 0 for large $n$, and so $\bar{\omega} \to \omega$ with probability 1.*

## 10 Empirical Results (continued)

### 10.1 Simulations

#### 10.1.1 Number of control units

With a large number of units, the risk of overfitting increases, which produces a high potential of increased estimation bias (Abadie et al., 2010). Here, we model the increase of the number of units with fixed $T_0 = 25$ and total $T = 125$. Large number of control units, $N$, has shown to be challenging for SC as it exacerbates the bias of the estimator (Abadie, 2021).

Figure 6 shows the effect of a growing number of units $N$ on algorithm performance. Given that EOpR also achieves low error at the post-intervention estimation at greater sizes of $N$, EOpR has the ability to reconstruct the true signal trend even with large control units and potentially noisy settings.

#### 10.1.2 Relative length of pre-intervention period

Consider when the number of pre-treatment units vary proportionally to $T$, fix the size of units $N$ and the total size of time $T$. We vary the time of intervention $T_0$ between 10% to 90% of the entire time $T$.

Figure 7 shows the effect of different pre-treatment lengths on the algorithm's ability to estimate, with $N = 50$ and $T = 200$. When having either a small or large number of pre-treatment periods relative to $T$, EOpR recovers the original treated signal with the smallest error compared to other algorithms.

Additionally, at lengths of 40% and onwards, the number of pre-treament vectors are greater than the size of $N$, $(N \ll T_0)$—a common setting adapted by Abadie et al. (2010)—EOpR still maintains a lower bias in the post-intervention with consistent low training error.

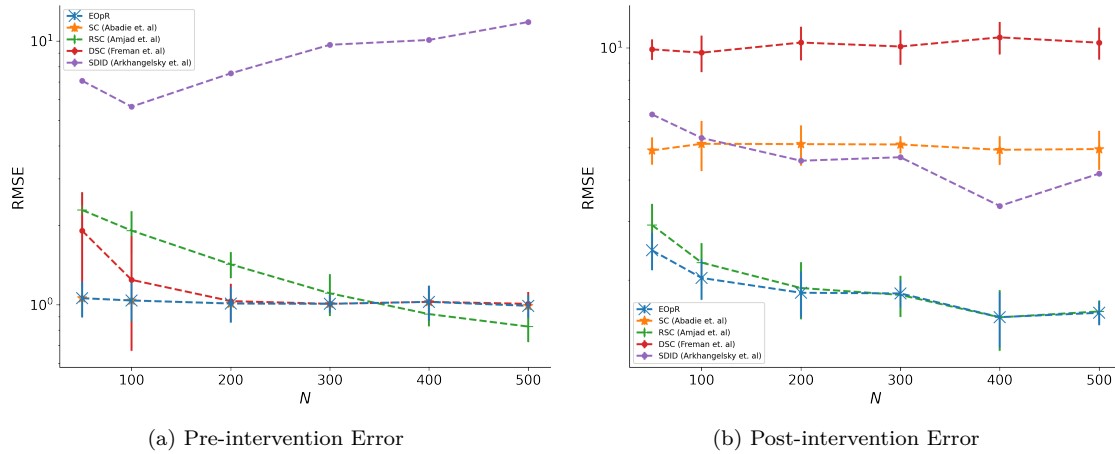

(a) Pre-intervention Error          (b) Post-intervention Error

Figure 6: Growing size of $N$, with $T_0 = 25$, and post-intervention $T = 100$

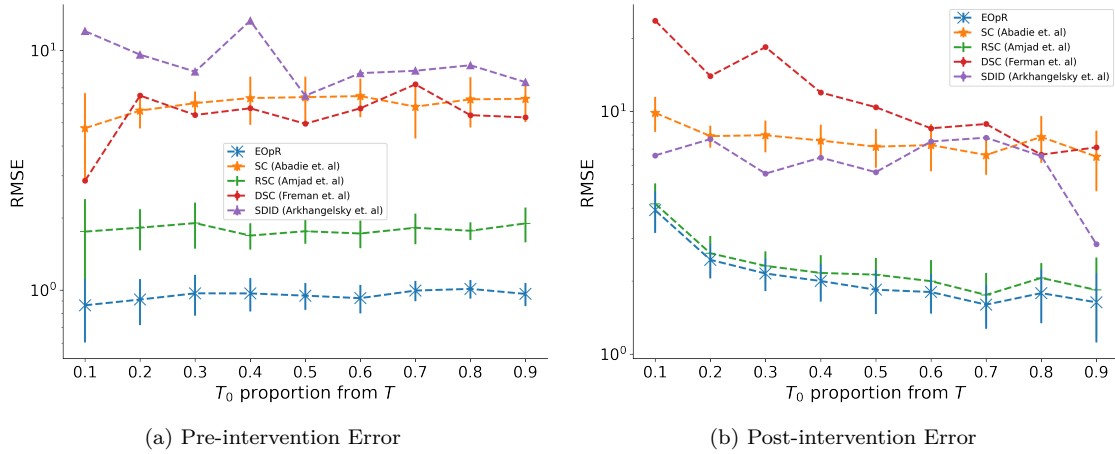

(a) Pre-intervention Error          (b) Post-intervention Error

Figure 7: Percentage of $T_0$ length of the total $T$, with $N = 50$, and total $T = 200$

### 10.1.3 Alternative (non-smooth) Simulations

We also consider a slightly different generating process, where we introduce non-smoothness, with heavy-tailed noise to create outliers and break the boundedness assumption. Therefore, we keep the same setup as in equation 14 but $\epsilon_{it} \sim t(3)$ belongs to a t-distribution with 3 degrees of freedom. The results are shown in Figure 8, which show that even with the existing of outliers, EOpR performs better than other baselines. We also consider another non-smoothness simulations with adding a third variable such that the sturcture becomes less compressible, e.g. $f(\theta, \rho) = \theta + \rho + \omega$. Results in 9 shows EOpR outperforms baseliens in most settings.

### 10.2 Additional Empirical Simulations

We consider an empirical simulation experiment derived from Arkhangelsky et al. (2019), using the Penn World Table Dataset. The dataset contains observations on annual GDP for 111 countries for 48 consecutive years. They fit the data into a rank-4 model then generate the treatment through a logistic model. They

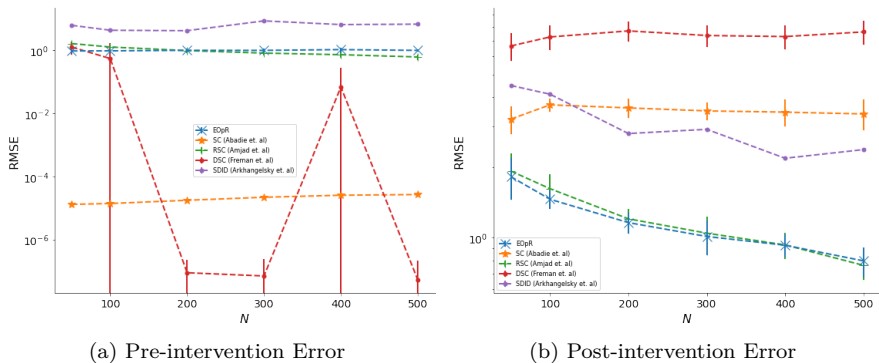

(a) Pre-intervention Error          (b) Post-intervention Error

Figure 8: Increasing $N$ under non-smooth with heavy-tailed noise setting

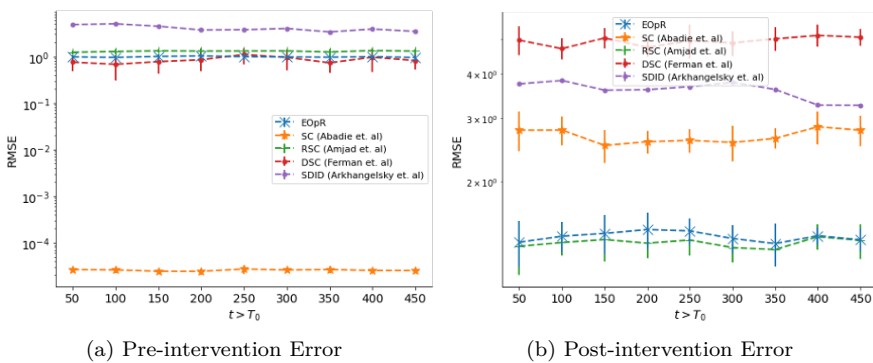

(a) Pre-intervention Error          (b) Post-intervention Error

Figure 9: Increasing $N$ under non-smooth setting with a third variable, e.g. $f(\theta, \rho) = \theta + \rho + \omega$.

ensure the assignment mechanism is correlated with observed unit characteristics, mimicking non-random treatment assignment in real-world scenarios.

We select log(real GDP) as the primary outcome. For the treatment assignment, we consider three choices, a completely random treatment assignment, non-uniform random assignment which is based on democracy, and based on education indicators.

Simulation results in Table 2 show that EOpR performs well under all three settings. The non-uniform random assignment setting represents real-world settings in which treatment assignment correlates with systematic effects. However, completely random assignment introduces zero bias, with low correlation within data. Having methods e.g., EOpR and RSC performing well under the non-uniform methods indicate that they are robust against data correlations.

Table 2: Simulation results based on the Penn World Table Dataset. Simulation setting based on (Arkhangelsky et al., 2019). All results are based on 100 simulations. First number is RMSE and second number between brackets is the standard deviation

|  | post-intervention Error | | | | | |
|---|---|---|---|---|---|---|
|  | Democracy | | Education | | Random | |
| EOpR | **0.154** | (0.099) | **0.125** | (0.064) | **0.179** | (0.148) |
| SC (Abadie et al.) | 0.162 | (0.091) | 0.157 | (0.096) | 1.339 | (0.997) |
| RSC (Amjad et al.) | 0.175 | (0.133) | 0.132 | (0.072) | 0.202 | (0.166) |
| DSC (Ferman et al.) | 0.448 | (0.227) | 0.434 | (0.210) | 1.266 | (0.868) |
| SDID (Arkhangelsky et. al) | 0.793 | (0.473) | 1.142 | (0.393) | 0.730 | (0.486) |

## 10.3 Real-world Experiment

### 10.3.1 California Proposition 99

The objective of this case study is to investigate the anti-tobacco legislation, Proposition 99, on the per-capita cigarette consumption in California in comparison to other states in the United States. The legislation took place in 1970. Without such legislation, the consumption of cigarettes in California would not have decreased (Abadie et al., 2010).

**Results.** Figure 10 shows the actual trajectory of California cigarette consumption in black. Our method recovers the estimated signal better than other methods with an adequate fit on the pre-intervention outcome, and also it derives the worst-case estimates.

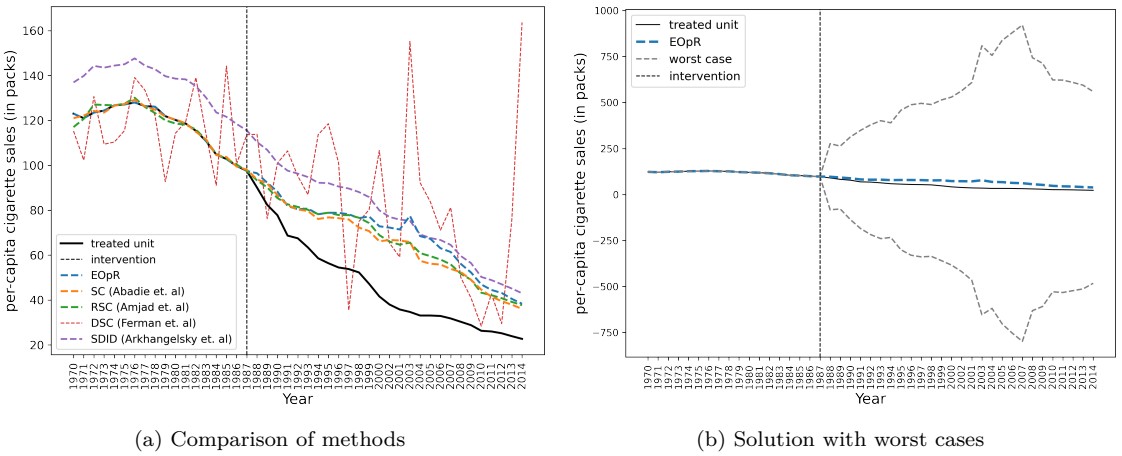

(a) Comparison of methods                    (b) Solution with worst cases

Figure 10: Trends in per-capita cigarette sales between California vs. synthetic California

**Placebo Tests.** We apply the same placebo test to the California case study. Abadie et al. (2010) excluded 12 regions, but we keep all of them. Figure 11 shows the divergence between estimations and observations of all regions with solid black line for California. This shows a similar observation as in Abadie et al. (2010).

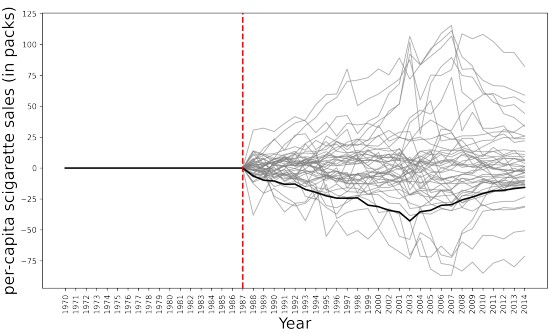

Figure 11: Placebo tests including all regions

### 10.3.2 Additional Placebo Tests

Here we make a comparison between placebo effects estimated by EOpR, SC, and RSC for the Basque experiment. We see that in all three algorithms, the counterfactual estimation of the treated unit diverges the most from its original vector. This indicates that the estimations are as expected by the those algorithms (See Figure 12).

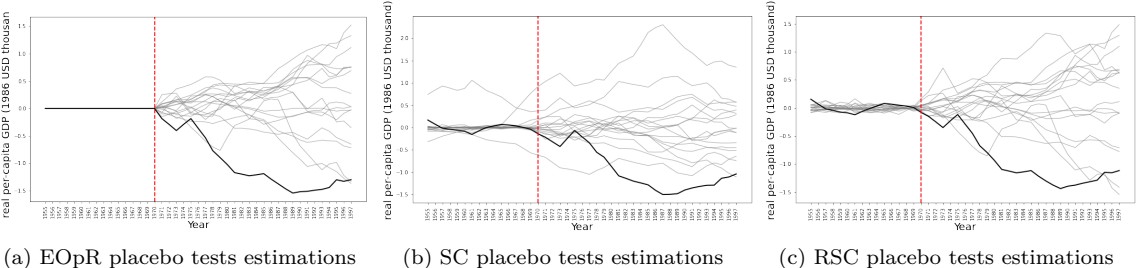

(a) EOpR placebo tests estimations     (b) SC placebo tests estimations     (c) RSC placebo tests estimations

Figure 12: Placebo tests comparison between methods

## 10.4 Ablation Study

We empirically study the effect of the added perturbation $\lambda$ to the covariate matrix $\boldsymbol{X}\boldsymbol{X}^\top$. This small perturbation is added to ensure $\boldsymbol{Q}$ is positive definite in equation 4, following the definition of ellipsoids in equation 1. When $\lambda = 0$, we see the resulting predictions suffer drastically, deviating from expected possible outcomes (Figure 13a). Using an appropriate $\lambda$ as in Figure 13b, balances the model complexity which helps safeguard the algorithm from potentially underfitting the training data, and producing a biased estimate.

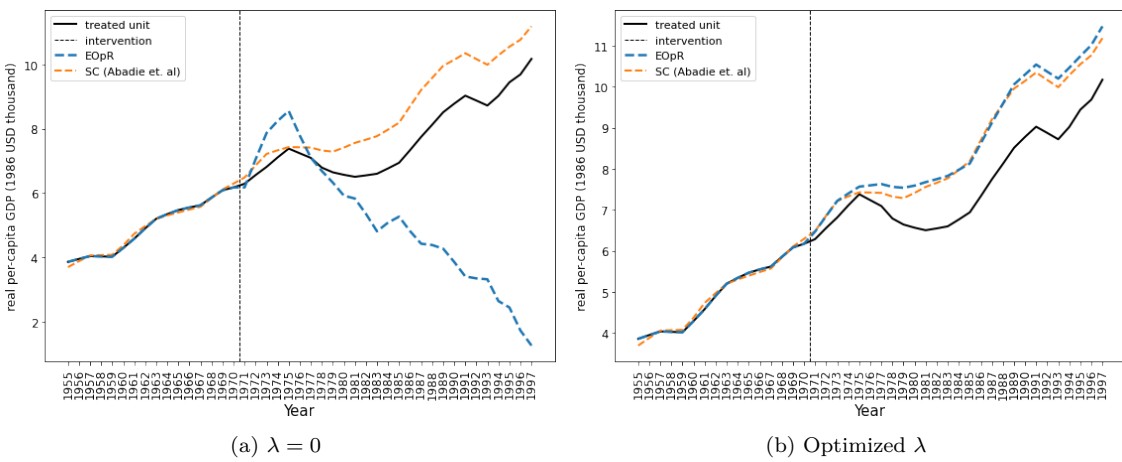

(a) $\lambda = 0$         (b) Optimized $\lambda$

Figure 13: Impact of adding $\lambda$ to the covariance matrix $\boldsymbol{X}\boldsymbol{X}^\top$ for Basque case study

A similar observation is seen for simulated data, fixing $T_0 = 100$ and $T_1 = 10$ and varying $N$, and for each combination we generate 10 simulations and average the resulting error scores. Figure 14 shows that with optimized $\lambda$ the bias and variance are consistently low, whereas estimations with non-optimized $\lambda$, i.e. $\lambda = 0$ estimations suffer from high error.

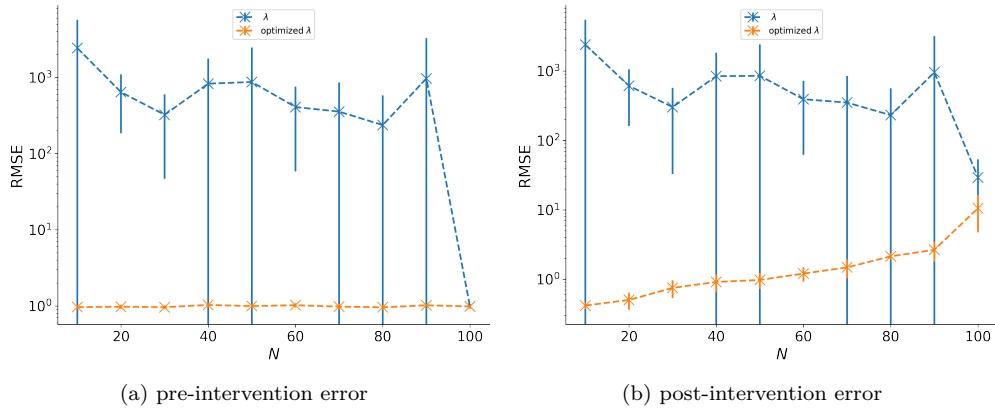

(a) pre-intervention error

(b) post-intervention error

Figure 14: Impact of adding $\lambda$ to the covariance matrix $\boldsymbol{XX}^{\top}$ for synthetically-generated data, fixed $T_0 = 100$ and $T_1 = 10$ with varying $N$.

## 10.5 COVID-19 Experiment (continued)

**Post-treatment length = 60** Figure 15 below shows the estimated trend of daily average cases of New York in comparison to the actual New York trend (treated unit in the figure). The estimation starts at $T_0 \in \{5, 10, 15, 20, 25, 30\}$. The worst-case estimates are also plotted.

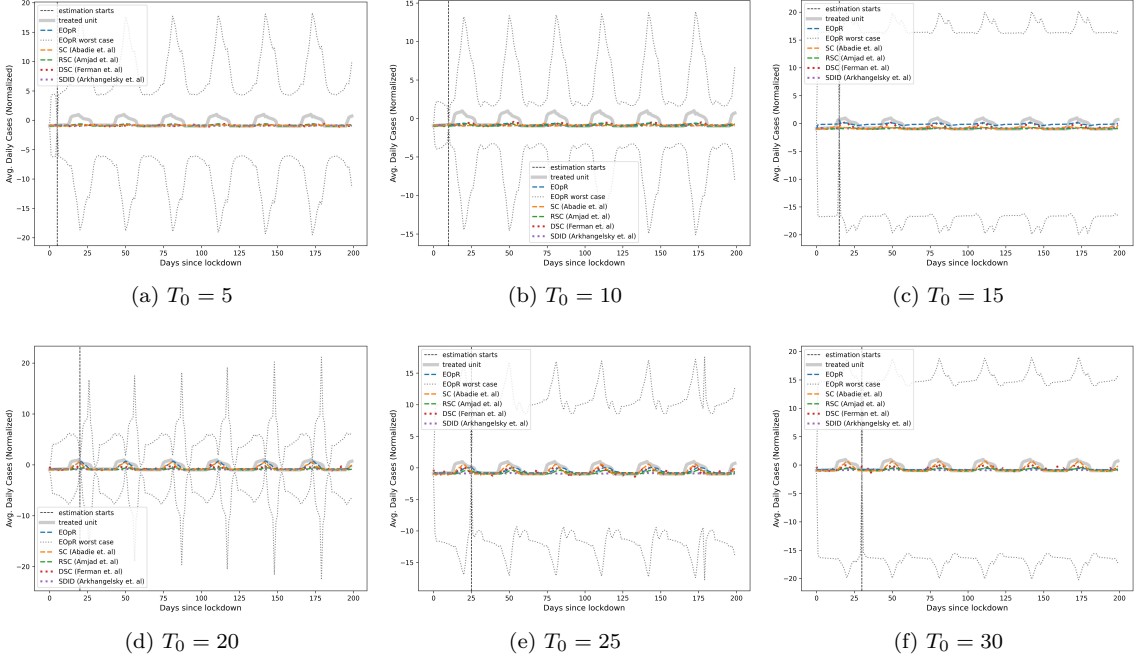

(a) $T_0 = 5$     (b) $T_0 = 10$     (c) $T_0 = 15$

(d) $T_0 = 20$     (e) $T_0 = 25$     (f) $T_0 = 30$

Figure 15: Comparisons of methods for estimating New York trend

