# OpenReview forum: "Ellipsoidal Optimal Recovery: A Minimax Approach to Robust Counterfactual Estimation"
_TMLR — Rejected by TMLR_

### Review · Reviewer_KDvo · 2025-03-27

**Summary Of Contributions:**

This paper presents a min-max geometric approach to recover unobserved outcomes of interest by borrowing idea from optimal recovery, the idea is to minimise the worst case reconstruction. The authors compare their methods with the long, rich synthetic control literature and demonstrate the effectiveness of their methods with synthetic and real-world datasets in comparison with other baseline methods.

**Audience:**

Yes

**Broader Impact Concerns:**

N / A

**Claims And Evidence:**

Yes

**Requested Changes:**

The first requested changes is weakness 1 I mentioned earlier, which is very critical for the acceptance of this work in my opinion. weakness 2 is not a must but will significantly improve the scientific aspect of this paper. I would like the authors to at least mention this issue as it is highly related to weakness 1.

The last comments is more like a bonus point for the sack of completeness in the literature and make this work more applicable in practice.

**Strengths And Weaknesses:**

**Strength**

1. The paper is generally well-written and with good flow.
2. The paper presents an interesting perspective which focus on "robustness of counterfactual estimation in worst case scenario", the proposed methods is clearly motivated and a good amount of contribution is made as this seems not been discussed in terms of synthetic control setting.


**Weakness**

1. The assumption of "ellipsoidal signal class" is a bit strong and not well justified, not sure I am very convinced about this on page 2 " ...which is a common assumption in economic ....". The authors are advised to have a section dedicate to this discuss on the assumption (as it is the major assumption here) and why/when this is applicable. Instead of claiming this is a general solution for many problem, it will be of great interest to the community to know when/what type of problem this min-max with "ellipsoidal signal class" assumption works well.

2. The experiments shows the good performance of this method, but I wondering if the authors are able to provide experiments that give negative results (for instance min-max on worst scenario would not make sense if the mean estimation is more favourable, e.g. some cases, high risk but also high reward). This may be possible by using a difference data generation process for the simulated experiments and this gives a hint on when/where the proposed method is losing.

**Other Comments**

One of the main concern in causal inference is how to estimate under unobserved confounders, some previous work in the literature such as "Agarwal, Anish, Devavrat Shah, and Dennis Shen. "Synthetic interventions." arXiv preprint arXiv:2006.07691 (2020)." solve this issue by thinking latent factors as a way of mediator the impact of these latent confounders. How will this min-max method tap into this setting, could you add this bit in the paper discussion/limitation?

---

> ### Author Response · Authors · 2025-05-24
> **Response to KDvo**
>
> We thank the reviewer for finding our paper well-written and with good flow.
>
> We copy-paste reviews and respond to them point-by-point.
>
> **1. The assumption of "ellipsoidal signal class" is a bit strong and not well justified, ... to know when/what type of problem this min-max with "ellipsoidal signal class" assumption works well.**
>
> We appreciate the reviewer suggestion to emphasize the justification behind the assumption that ellipsoidal optimal recovery is a potential solution for econometrics problems. To justify our assumption and proposed method, _we add several paragraphs in the introduction section in a new section labeled “Why Ellipsoids?” as follows._
>
> “We posit that phenomena under study have signals that belong to a particular ellipsoidal signal class, i.e. a quadratic function class (a particular type of convex function class). This is a common assumption in nonparametric econometrics in which signal classes are constructed to characterize bounded smoothness or shape restrictions  (Armstrong & Kolesár, 2018; Cheng et al., 1997).  It is a valid assumption since nearly all real-world societal signals are smooth. Moreover in many real-world scenarios, data is also naturally bounded, e.g. income data,  test scores, gross domestic product (GDP), and others. Such quadratic signal class assumptions also abound in numerous applications in science and engineering (Lorentz et al., 1996) . Therefore, convex functions are better to adapt to the bounded data structure and produce estimates that are accurate with minimal bias (Cheng et al., 1997).
>
> Let us separately note that Armstrong & Kolesár (2018) show that assuming convex function classes (such as ellipsoids) enables optimal inference in linear regression. Indeed, the degree of smoothness of a function directly governs how well it is approximated, e.g. using low-degree polynomial reconstructions (Lorentz et al., 1996). Further, Armstrong & Kolesár (2018) derive minimax bounds and confidence intervals over convex classes under shape or smoothness constraints. This jives with our motivations to consider ellipsoidal classes and obtain minimax estimates.
>
> In our setting, the ellipsoidal signal class captures the underlying geometric structure and dependencies in the data, enabling robust extrapolation of the treated unit even under a short pre-treatment period or noisy conditions. Note that our method does not aim to be universally applicable, but rather provides a minimax optimal solution for ellipsoidal classes, which is suitable for numerous practical scenarios especially in the noisy and low-sample regime.”
>
> We further discuss failure cases in the Discussion section in text, see Response to R8ch (Part 3), answer 5.2 https://openreview.net/forum?id=iVxlbNl8Ow&noteId=F29p3immcB
>
> **2. The experiments shows the good performance ..., provide experiments that give negative results ..."**
>
> We thank the reviewer for their suggestion regarding evaluating our method under alternative  data generating process. In our original data generating process, our generating process setup uses a smooth bounded logistic model with parameters drawn from uniform distribution combined with standard gaussian noise.
>
> As per the reviewer's suggestion, we consider a heavy-tailed noise $\epsilon \sim t(3)$ to break the bounded ellipsoidal shape and create outliers.  We see in figures that EOpR did not perform well in pre-intervention error unlike in the original setting.
>
> Another alternative generating process is to add a third variable to make structure less compressible, e.g. $f(\theta, \rho) = \theta + \rho + \omega$
>
> _Those new experiments were added to Appendix under “Simulated Experiments”._
>
> **Other Comments**
>
> **One of the main concern in causal inference is how to estimate under unobserved confounders, ... How will this min-max method tap into this setting, could you add this bit in the paper discussion/limitation?**
>
> Yes, we agree with the reviewer that previous works considered latent factors and low-rank analysis to learn signal information and reduce noise. We follow a similar approach by learning the ellipsoid, which learns the signal class. Then, having the learned ellipse\oid allows us to arrive at a min-max solution. We discuss this and added several paragraphs in text to highlight this point.
>
> _In section 1, “Algorithmic” paragraph we mention:_ “Unlike statistical approaches (Abadie & Gardeazabal, 2003; Abadie et al.,2010) or low-rank matrix decomposition approaches (Amjad et al., 2018; 2019; Athey et al., 2021) which minimize the average error, our approach minimizes the maximum error over the known samples, which is intended to make EOpR a robust algorithm, among other properties. It has previously been proven that minimizing the maximum error produces robust estimators with small bias (Kassam & Poor, 1985; Muresan & Parks, 2005)”.
>
> In section 1, see the newly added paragraph “Why Ellipsoids?” for additional emphasis on our min-max ellipsoidal approach.

---

> ### Author Response · Authors · 2025-06-13
> **Response to reviewer KDvo comments**
>
> We thank the reviewer for finding our responses to be detailed and clear. We respond to the raised comments below:
>
> 1. We uploaded a newer version incorporating all reviewers comments in which additions are colored in red.
>
> 2. **Comment 1: "On the data generation process and the negative results: while I find these contributions valuable, I am not sure they are clearly described or sufficiently discussed in the current manuscript. Could you please expand on this aspect, ideally with a dedicated discussion section? It would also be helpful if you could point me to the specific figures or sections where these results currently appear."**
>
> - For the new data generating process, please see in the newly uploaded text, in appendix, page 19, section 10.1.3., where we presented the reviewer's suggestion of trying an alternative data generating process giving negative results.
>
> - Further, we thoroughly discuss successful and failing scenarios of our method in the Discussion section, please see page 12.
>
> 3. **Comment 2: As for the point about unobserved confounders, I still find the explanation a bit unclear. While I agree that latent variables are a broader concept, they are not always confounders. The term "latent variable" is often used loosely, and I would appreciate it if you could clarify the distinction in your context and elaborate more precisely on your assumptions.**
>
> We agree with the reviewer that latent variables are not necessarily confounding factors, as they are not assumed to influence both treatment assignment and potential outcomes. Instead, latent variables are assumed to influence the outcome process, such that they capture the time and unit specification of an outcome. We use the term latent factors to refer to low-dimensional factors that generate the time series trajectories of both treated and control units. This aligns with standard assumptions in the synthetic control literature and factor modelling (Athey, 2021). Such formulation subsumes popular econometric factor models presented in (Abadie, 2010), and (Abadie et al., 2019).
>
> _To clarify, we include the following in our text in section 3:_
>
> “Notably, we use the term latent factors to refer to low-dimensional factors that generate the time series trajectories of both treated and control units. This aligns with standard assumptions in the synthetic control literature and factor modeling (Athey, et al., 2021; Abadie, et al., 2010; Amjad et al., 2018). The latent space is assumed to influence the outcome process, unlike confounding factors that are assumed to influence the treatment assignment process. We defer studying confounding factors to future work.”

---

### Review · Reviewer_PY1b · 2025-04-02

**Summary Of Contributions:**

The paper presents a novel method to estimate the effect of an intervention on time series data.
The algorithm, called EOpR, finds the center of the intersection between an ellipsoid fitting the data and a hyperplane, and uses this as the estimate for the unobserved values of the time series without the intervention, inspired by "synthetic control".
Results on synthetic and real-world data are presented.

**Audience:**

Yes

**Broader Impact Concerns:**

None.

**Claims And Evidence:**

No

**Requested Changes:**

As per the list above, the authors should invest a significant effort in improving the presentation of the paper in order to make it accessible to an audience outside the specific area.
This means improving notations, justifying assumptions, accurately describing relevant background and previous work.

**Strengths And Weaknesses:**

As someone not working on the subject, I found the paper extremely hard to read and understand.

1) As far as I understand, the work is inspired by the "synthetic control" family of methods presented in section 2.2. However, these methods are only briefly outlined: no mathematical details are provided, and no clear indication of the technical differences compared to the proposed approach. This makes it quite hard to assess the technical merit of the work, as well as to understand the experimental results.

2) As far as I understand, the paper is based on the assumption that control vectors belong to an ellipsoid, and that the treated unit whose untreated data needs to be recovered lies at the intersection between a hyper-plane and this ellipsoid. As someone outside the area, it is unclear to me how standard these assumptions are. I believe they should be more carefully justified. Why would pre-interventions of control units belong to a hyper-plane? Also, aren't pre-interventions lower-dimensional vectors? I feel like I am clearly missing something here, and a more thorough explanation and presentation would definitely help.

3) I think the paper has several typos and notational issues. For instance, at the beginning of page 4, I am not sure I understand what $\eta_i \psi_t$ means. These are vectors of different lengths: is this an inner product, a Hadamard product, a product between scalars? Related to this, it seems that subscripts are used to access vector coordinates, but then why is $s_{1t}$ a T-dimensional vector (end of page 3)? $B_r$ and $B_c$ appear to look the same to me, before equation (14). Terms are used without definition: what are unbiasedness and consistency?

4) The proposed algorithm would seem to display some similarities with PCA, at least when learning the ellipsoid. I imagine these sorts of ideas must have been also employed by previous work.

5) Given that it is integral part of the proposed algorithm, the Riesz representation theorem should be presented and explained.

6) What precisely is a worst-case estimate in this context and why is it useful? From Figure 4b, the provided estimates would appear to be extremely loose.

7) The empirical results are not fully convincing to me, or at least they are not appropriately commented upon. Why is the EOpR pre-intervention error significantly larger than for SC on Figure 3? In figure 5, the placebo estimates do not look particularly accurate to me. I imagine it could also be the case for competing methods, but then the authors should provide similar plots for the baselines and explain the relative differences. Also, what is the post-intervantion RMSE computed on for Table 1? What is the ground truth?

---

> ### Author Response · Authors · 2025-05-24
> **Response to PY1b (Part 1) [Points 1,2]**
>
> We thank the reviewer for finding our work novel and our method and results to be well-presented.
>
> **1. As far as I understand, the work is inspired by the "synthetic control" family of methods presented in section 2.2. However, these methods are only briefly outlined: ... and no clear indication of the technical differences compared to the proposed approach ...**
>
> We thank the reviewer for raising this essential point. _Indeed, our method is inspired by the synthetic control family, but diverges in assumptions and technical formulation._ There are several differences between EOpR and traditional synthetic control (and its extensions).
>
> _We further added the following technical differences in the main text in the Discussion section._
>
> "First, synthetic control (SC) estimates a counterfactual treated unit by linearly interpolating on control units. Contrarily, EOpR estimates a counterfactual treated unit by extrapolating on the covariance of control units. Having the covariance instead of raw control units captures inter-correlation between data points, which makes it more robust for estimation in noisy settings.
>
> Second, SC constrains the weights to sum to unity so they lie on the probability simplex; other SC variants such as DSC relax this assumption with unconstrained weights, whereas RSC allows weights to be negative using ($L_1$ or $L_2$) regularization. On the other hand, EOpR uses the Riesz representation theorem to construct the weights with an inner product of covariance matrices. In other words, the weights are a linear combination of representors, which are obtained by the Riesz representation theorem. The representors are used to represent inner products of covariance matrix.
>
> Third, stemming from the Riesz representation theorem, EOpR derives worst-case estimates, which are two vectors on the farthest edges of the ellipsoid, in the direction of the center. The worst-case estimates can be thought of as bounding the estimation of the treated unit.  There is no such worst-case characterization that comes from traditional SC or its variants."
>
> **2. As far as I understand, the paper is based on the assumption that control vectors belong to an ellipsoid, and that the treated unit whose untreated data needs to be recovered lies at the intersection between a hyper-plane and this ellipsoid. ... it is unclear to me how standard these assumptions are.  ...**
>
> Please see response to Reviewer KDvo regarding how standard the assumption of smooth signal classes and other quadratic (ellipsoidal) signal classes are in econometrics and beyond.
>
> Please also further note that in panel data, pre-intervention data are part of time series that are sampled at given times.  Time samples are directly and naturally linear functionals of the underlying phenomenon.  When considering several linear functionals, they define a hyperplane.  This hyperplane is a lower-dimensional manifold in the higher-dimensional space of all time points (both pre-intervention and post-intervention) that the ellipsoid lies in.  When a (pre-intervention) hyperplane cuts an ellipsoid, one gets a lower-dimensional ellipsoid (the post-intervention ellipsoid).  The Chebyshev center of the post-intervention ellipsoid is the estimate.
>
> Recall that our goal is to find a vector that belongs to the data, and in the same way, completes the missing parts of the data. That vector should be very close to other vectors, and that’s the recovery part. Based on the Chebyshev theorem, an intersection of two shapes (ellipsoid and hyperplane) has a center that is the minimal farthest vector from all other points. Such a vector would be the (best) vector which is closest to all other vectors with least min-max error.
>
> Based on the above, we wanted to have two shapes. According to the optimal recovery theory from Muresan and Parks (2002), we want to find an ellipsoid that is representative of the control units (training data). Given that the ellipsoid may be stretched in a specific direction more than others, this property gives the ability to select vectors in the direction of the most stretch, such that they closely capture most information in the data. Such vectors can be viewed as the principal components of the data. That’s why we assume control units to belong to an ellipsoid.
>
> For the second shape, we assert the pre-intervention units to belong to a hyperplane. (Recall that control units are divided into two sets: pre-intervention and post-intervention units). To justify our assumption, recall from the Chebyshev theorem that the two planes must intersect. Second, given that the representation theorem works for Hilbert spaces, a hyperplane allows us to obtain representers from the pre-intervention data, and so to get the weights explaining the relationship between pre-intervention vectors of control units and the pre-intervention vector of the treated unit. Therefore, this allows us to extrapolate for the post-intervention treated unit.

---

> ### Author Response · Authors · 2025-05-24
> **Respons to PY1b (Part 2) [Poitns 3-5]**
>
> **3. I think the paper has several typos and notational issues. For instance, at the beginning of page 4, I am not sure I understand what \nu_i\ means. These are vectors of different lengths: is this an inner product, a Hadamard product, a product between scalars? ... Terms are used without definition: what are unbiasedness and consistency?**
>
> We thank the reviewer for helping us identify issues to be fixed.
>
> We fix detected typos and notational issues. Specifically, on page 4, we intended to say that $s_{it}$ is a function of both the unit vector $\nu_i \in R^N$ and the time vector $\psi_t \in R^T$. To make it clearer, we rewrite it as following:
> 					$$s_{it} = f(\nu_i, \psi_t)$$
> where $f(\cdot)$ is a latent function that captures the model relationship, following the common factor model in econometrics presented in Abaide et al. (2010).
>
> For the subscripts, yes they are used to access vector coordinates. For $s_{1t}$, it is the treated vector for all time $t$. To distinguish between other control units and the treated unit we fix $i=1$ to be for the treated unit, for all time, $t \leq T$.
>
> _To clarify, we rewrite the sentence in the manuscript as following:_ "For the treated unit of interest, we fix $i=1$ ($i\in \{1,\dots, N\}$) for time $t \leq T$, hence, let $s_{1t} \in \mathbb{R}^{T}$, for brevity, we use $s_1$ as well"
>
> For $B_r$ and $B_c$ before equation (14), we intended to have them similarly generated, to have them work as latent row and column parameters following Amjad et al. (2019).
>
> Thanks for pointing out about defining the terms of unbiasedness and consistency. _We write the sentence in the manuscript to clarify as following:_ “We say that the estimator is consistent when $\mathit{MSE}$ error converges to zero if $T$ grows without bound.”. For unbiasedness, we rewrite the sentence to emphasize the definition as follows: “An estimator is said to be unbiased when the estimated vector equals the true vector on average”.
>
>
> **4. The proposed algorithm would seem to display some similarities with PCA, at least when learning the ellipsoid. I imagine these sorts of ideas must have been also employed by previous work.**
>
> Yes, we agree with the reviewer that constructing the ellipsoid is similar to PCA. Specifically, the principal components of the data $A$ are in the direction of the most stretch of the ellipsoid $K$. The principal component vectors are the eigenvectors corresponding to the largest eigenvalues of $AA^{\top}$.
>
> Indeed, PCA and reduction analysis in general have been employed in previous work for causal inference, to estimate missing outcomes. Though learning the ellipsoid is similar to low-rank estimation, we also employ convex inner products to obtain representors and derive weights, which differs from methods in previous works.
>
> _To emphasize this note, we added a paragraph “Connection to PCA” in section 4.1 in text as following:_
> “Constructing the ellipsoid in some way is similar to principal component analysis (PCA). Specifically, the principal components of the data matrix $A$ are in the direction of the most stretch of the ellipsoid $K$. The principal component vectors are the eigenvectors corresponding to the largest eigenvalues of $AA^{\top}$. PCA have been employed in previous work for causal inference (Agarwal et al., 2020; 2023), to estimate missing outcomes. Though learning the ellipsoid is similar to low-rank estimation, however, we also employ convex inner products (as we see next) to obtain representers and derive weights, then learn a min-max optimal estimator which differs from previous works.”
>
> **5. Given that it is an integral part of the proposed algorithm, the Riesz representation theorem should be presented and explained.**
>
> We thank the reviewer for suggesting to thoroughly explain the Riesz representation theorem in the paper to make it more contained.
>
> _We add the following paragraph to the Appendix for further reading._
> 	"The Riesz representation theorem is fundamental in functional analysis, it represents continuous linear functionals as inner products with specific vectors in the space. It establishes a deep connection between Hilbert spaces and their duals.
> 	Let H be a Hilbert space over R and a function $\phi: H \to R$ is called a linear functional.
> Theorem (The Riesz representation theorem). For a continuous linear functional \phi on a Hilbert space H, there exists a unique u ∈ H such that $\phi(x) = <x,u>, x_i$ for all $x \in H$ . Furthermore, $||u||=||\phi||$"
>
> Further technical details in https://math.uchicago.edu/~may/REU2021/REUPapers/Adler.pdf

---

> ### Author Response · Authors · 2025-05-24
> **Response to PY1b (Part 3) [Points 6-7]**
>
> **6. What precisely is a worst-case estimate in this context and why is it useful? From Figure 4b, the provided estimates would appear to be extremely loose.**
>
> Based on the Chebychev theorem, the estimated vector of interest is the center of intersection, which has the minimum maximum error (least farthest vector from the unknown vector that we want to recover). The intersection is an ellipse, and an ellipse has two farthest points at opposite edges lying at the boundary. They bound the center point, and reflect the maximum possible deviation from mean estimation while remaining consistent with underlying model assumptions.
>
> These bounds show that the algorithm produces credible worst-case intervals rather than overfitting to noise, or producing spurious certainty. Its usefulness appears in noisy settings and policymaking under deep uncertainty, especially that decision makers need not only to understand expected effects but also what could go wrong.
> Regarding Figure 4b, we agree that it is important to evaluate whether the worst-case bounds are meaningful or overly conservative. Given the noise in historical data and the extrapolation beyond observed inputs, the difference between mean estimate and bounds is not abnormally large, about 20-25 units, which is within an order of magnitude, and reflects a reasonable level of uncertainty.
>
> _To avoid confusion, while describing results in Figure 4b, we further discuss the worst-case estimates as following:_
> “Given the noise inherited in historical data and the extrapolation beyond observed inputs, worst-case estimates may look loose, however, the difference between mean estimate and bounds is not abnormally large, about 20-25 units, which is within an order of magnitude, and reflects a reasonable level of uncertainty.”
>
> **7. The empirical results are not fully convincing to me, or at least they are not appropriately commented upon. Why is the EOpR pre-intervention error significantly larger than for SC on Figure 3? ...**
>
> We divide the reviewer’s comment into three parts, to answer each sub-question.
>
> **7.1. Why is the EOpR pre-intervention error significantly larger than for SC on Figure 3?**
>
> In Figure 3, the configuration is $N \ll T$, such that $N=100$ and $T$ grows to 450, which is the best configuration for SC to work with. SC is better at solving fat matrices where columns are larger than rows (Athey et. al, 2021). Therefore, this could be a reason that SC outperforms EOpR in pre-intervention error, however, it did not beat EOpR in the post-intervention.
>
> **7.2. In figure 5, the placebo estimates do not look particularly accurate to me. I imagine it could also be the case for competing methods, but then the authors should provide similar plots for the baselines and explain the relative differences.**
>
> Recall that the placebo test considers one control unit as a placebo treated unit at a time and apply the estimation algorithm. Since control units are assumed to not be affected by the examined intervention, one would expect that the estimated signal for the placebo unit does not diverge from its corresponding control unit. Further, the gaps between each placebo estimation and its corresponding control unit should be less divergent than the gap between the original treated unit and its estimation.
>
> In Figure 5, the most divergent line is the line corresponding to the actual treated unit of interest, while the rest of units have less divergence from their original values after being estimated. We provide similar plots for the RSC and SC methods, which we reproduce from their results in their papers. (See Appendix 10.3.2)
>
> **7.3. Also, what is the post-intervention RMSE computed on for Table 1? What is the ground truth?**
>
> In this experiment, we estimate the “factual” trend instead of the “counterfactual”, given that we already have the factual trend of New York as the post-intervention, which is the ground truth. Given the ground truth and the estimated trend by algorithms, we compute the error. This method of testing the reliability of estimation algorithms has been previously used in various works, e.g., Bayat et. al (2020).
>
> _To avoid confusion, we add this discussion to section 5.3.2 “COVID-19 in New York” in manuscript_

---

> > ### Comment · Reviewer_PY1b · 2025-06-12
> >
> > I thank the authors for their very detailed and kind response.
> > I agree with reviewer KDvo: the changes are heading in the right direction.
> > However, it would seem to me that the updated revision coincides with the original submission, and does not contain the authors' described changes.
> > I would hence urge the authors to update the manuscript in order to see the changes in full (perhaps, appropriately highlighted by a different text color).

---

> > > ### Author Response · Authors · 2025-06-12
> > > **Uploaded new version**
> > >
> > > We thank the reviewer for finding our response helpful and detailed.
> > >
> > > Please re-check our uploaded manuscript, with changes colored in red.

---

> > > > ### Comment · Reviewer_PY1b · 2025-06-17
> > > >
> > > > Thank you for uploading the manuscript.
> > > >
> > > > I confirm that the revision has increased the quality of the submission. However, my original concerns remain valid.
> > > >
> > > > I still believe that the quality of the presentation needs to be improved in order for the submission to be accessible to a wider audience. For instance, I appreciate that the authors have now more heavily cited work in econometrics to justify the ellipsoid assumption. But this still feels very strong given the complexity of the studied phenomena.
> > > > Pre-intervention samples are declared to be linear functionals of the full signals in section 4.2.1 without any prior mention.
> > > > Likely related to this, it is unclear to me how pre-intervention vectors, which belong to $\mathbb{R}^{T_0}$, can be mapped into $\mathbb{R}^{T}$.
> > > > Furthermore, the authors assume that the full true signal lies at the intersection between the ellipsoid containing all signals, $\mathcal{K}$ and the hyperplane of the pre-interventions of the control units, $\mathcal{H}$: while the assumption is clear for $\mathcal{K}$, I do not quite grasp why it would hold for $\mathcal{H}$, which (as far as I understand) only contains pre-interventions.
> > > > Finally, in equation (7), aren't $\Sigma \in \mathbb{R}^{T \times T}$, and $\hat{w} \in \mathbb{R}^{T_0}$?
> > > >
> > > > I also still believe the notation to be confusing: for instance using the subscript t to both mean a vector valid "for all t <= T" and the t-th entry of such vector (for instance, when defining $A^-$).
> > > >
> > > > I am also not fully convinced of the utility of the worst-case analysis as demonstrated in Figure 4, where I would disagree with the authors on the level of uncertainty being reasonable. I would also urge the authors to either compare against the balancing weights methods that are now discussed in section 2.2, or to provide an explanation as to why this cannot be done.

---

> ### Author Response · Authors · 2025-07-17
> **Response to PY1b second feedback (Part 1)**
>
> We thank the reviewer for finding our manuscript has increased in quality. We reply to the comment point-by-point.
>
> **_(Newly uploaded manuscript includes following changes colored in blue)_**
>
> **1.But this still feels very strong given the complexity of the studied phenomena. Pre-intervention samples are declared to be linear functionals of the full signals in section 4.2.1 without any prior mention**
>
> Please note that samples of a signal are always linear functionals of that signal, i.e. they satisfy the superposition and homogeneity properties. As such, we agree the wording as “are declared” was not the best and perhaps not the best placed in the flow.  We thank the reviewer for raising this concern. The sentence currently appears in section 4.2.1 in the context of justifying the invocation of the Riesz representation theorem.
>
> Our intended meaning is that we observe partial trajectories, i.e. the pre-intervention (specifically, the first T0 entries) of the treated unit's full latent signal vector, which are naturally linear functionals. When applying the Riesz representation theorem, this structure justifies the existence of representers — dual vectors that uniquely correspond to evaluations of the signal in the observed time indices under the inner product induced by the covariance matrix. Recall equations (5)-(6).
>
> _To make it clearer, we briefly **introduce linear functionals**, in an earlier paragraph, in section 4 as following:_
>
> “In this framework, the known values, such as sampled pixels or pre-intervention observations, are linear functionals. Linear functionals refer to any operation that maps a signal (image patch) to a scalar via linear mappings, e.g. derivatives, averages (Muresan et al., 2001). The representers, derived via the Riesz representation theorem, then map these known linear measurements back into the full signal space, allowing reconstruction imposing the structure of observed data.”
>
> _Further, to avoid confusion, we rewrite the sentence in **section 4.2.1**., by elaborating more on the intuition of representers and linear functionals as following:_
>
> “We can think of representers, in geometric terms, as the vectors passing through $Q$ in the direction of the ellipsoid. The time samples of the pre-intervention period are, by construction, linear functionals of the full signals. Since we observe the first $T_0$ time samples of the pre-intervention period, we consider them as fixed coordinate projections of the full signal vector. These projections define an affine constraint, and then the Riesz representation theorem guarantees that there exists a unique set of vectors---the representers—that map these constraints into the signal space via the induced inner product. (Further details on the Riesz representation theorem are in Appendix 8.1)”.
>
> **2. Likely related to this, it is unclear to me how pre-intervention vectors, which belong to RT0, can be mapped into RT.**
>
> Given representers, we can map pre-intervention vectors to $R^T$. Recall equation (7), given representers and pre-intervention matrix $\Sigma^{-} \in R^{T \times T_0}$, this returns a trajectory belonging to R^T. We believe our added explanations on the mapping of pre-intervention vectors in section 4.2.1 (above) resolves this concern.
>
> **3. Furthermore, the authors assume that the full true signal lies at the intersection between the ellipsoid containing all signals, K and the hyperplane of the pre-interventions of the control units, H: while the assumption is clear for K, I do not quite grasp why it would hold for H, which (as far as I understand) only contains pre-interventions.**
>
> Recall that the hyperplane is a lower-dimensional manifold in the higher-dimensional space of all time points (both pre-intervention and post-intervention) that the ellipsoid lies in. Technically, we consider $\Sigma$ to be the hyperplane and K to be the representer vectors in \Phi passing through Q (Eq. 5). The representers in K yield a feasible set of signals whose projection onto the first T0 time steps match the observed pre-intervention values. Therefore, the intersection of K and H is an induced subset that contains full signals that are consistent with the prior covariance structure (of ellipsoid K) and match the observed pre-intervention trajectory (by projection of representers).
>
> _To clarify, we add the following to section 4.2.2 in the manuscript_
>
> “Geometrically, we consider $\Sigma^{-} \in R^{T \times T_0}$ to be in the hyperplane H and K to be the set of representers vectors in \Phi (Eq. 5). The representers in K yield a feasible set of signals whose projection onto the pre-intervention vectors match the observed pre-intervention values. Therefore, the intersection of K and H creates an induced subset C that contains full signals that are consistent with the prior covariance structure (of ellipsoid K) and match the observed pre-intervention trajectory (by projection of representers).”

---

> ### Author Response · Authors · 2025-07-17
> **Response to PY1b second feedback (Part 2)**
>
> **_(Newly uploaded manuscript includes following changes colored in blue)_**
>
> **4. in equation (7), aren't Σ∈RT×T, and w^∈RT0?**
>
> We thank the reviewer for pointing this out. We intended to write \Sigma^{-}, which belongs to R^{T x T_0}, and \hat{w} \in R^{T0 x 1}.
> We fix equation 7 accordingly in the manuscript as following: “\hat{a}_1 = \mat{\Sigma^{-}}^{\top} \hat{w}”
>
> **5. I also still believe the notation to be confusing: for instance using the subscript t to both mean a vector valid "for all t <= T" and the t-th entry of such vector (for instance, when defining A−).**
>
> To clarify, the subscript $t$ is used to index a specific time period of a vector or matrix, but it is never intended to refer to the full time range unless made explicit (e.g., through notation like $t \forall t \leq T$). However, the negative sign $(-)$ used as superscript is made to reflect the range for all $t <= T_0$, used to index pre-intervention period of matrices and vectors, which is a standard notion used by Amjad et al., (2018).
>
> _We add the following note to the text in section 3:_
>
> “(Note that the subscript $t$ is used to index a specific time period of a vector or matrix, while the negative sign $(-)$ is used as superscript to reflect the range for all $t \leq T_0$, used to index pre-intervention period of matrices and vectors, similarly the positive sign $(+)$ reflects the post-intervention range for all $t > T_0$).”
>
> **6. I would also urge the authors to either compare against the balancing weights methods that are now discussed in section 2.2, or to provide an explanation as to why this cannot be done.**
>
> Balancing weights methods match the weights between control units and the treated unit in pre-intervention. They can be quadratic, linear, or any other type of function. Please note that we are already comparing against synthetic control and synthetic difference-in-differences, which are types of balancing weights methods.
>
> Balancing weight methods do not necessarily retrieve uncertainty measures for their estimates, which is why we cannot compare our worst-case estimates against any other. Our method, by construction of the ellipsoid, returns worst-case vectors for the estimate. Recall from section 4.3 that worst-case estimates mainly depend on data, given \Sigma and Q. Therefore, the uncertainty might be affected by the noise in data. For example if the amount of noise is large, we may expect a wider range of worst-case intervals.
>
> _We add the following to **section 4.3 on worst-case estimates:**_
>
> “In situations of deep uncertainty (Lampert et al., 2006), it is important to acknowledge the most severe possible outcomes that could occur for a given policy. In deep uncertainty, one is quite concerned about distinguishing what is possible from what is impossible, especially with limited knowledge about future conditions.”
>
> _Also, to avoid confusion, we add to section 5.3 the following note:_
>
> “Please note that our method returns worst-case estimates, whereas baseline methods consider statistical notions of uncertainty. Therefore they are not directly comparable.”

---

### Review · Reviewer_R8ch · 2025-05-09

**Summary Of Contributions:**

This paper is concerned with causal effect estimation in panel data. The setup is focused on the setting where pre-treatment control units and the pre-treatment values of the treated unit are used to estimate weights for the control units which are then used to impute the values of the missing potential outcome for the treated units. The authors propose an approach that replaces the typical regression or balancing weight approach with an approach that (1) first takes an ellipsoid given by taking $AA^\top + \lambda \mathbb{I}$, where $\lambda \geq 0$ is a ridge like parameter, (2) considering the hyperplane that intersects the ellipsoid defined by the treated unit, and (3) using the Chebyshev center to define an estimator which extrapolates to the missing potential outcomes. The authors do a small amount of theoretical characterization and provide a small set of demonstrations on well-known datasets.

**Audience:**

Yes

**Claims And Evidence:**

No

**Requested Changes:**

Largely, see above on the list of weaknesses.
To summarize:
1. Please add details in (a) the method definition, (b) related work, (c) assumptions, and (d) the theoretical framing.
2. Please explicitly consider $\lambda$ in derivations
3. Please provide empirical simulation that clearly outlines the performance of the proposed method.

Finally, it would be useful, though not strictly required, if the authors compared the proposed approach to an unconstrained balancing weight estimator with a ridge penalty, e.g. $\min_{w} || Aw - a+||^2 + \lambda ||w||^2$. This is a commonly used estimator, but see the following for a useful discussion: Bruns-Smith, David, et al. "Augmented balancing weights as linear regression." Journal of the Royal Statistical Society Series B: Statistical Methodology (2025).

**Strengths And Weaknesses:**

Strengths:
1. I found the framing around using an enclosing ellipsoid an interesting approach and believe it may have interesting applications.
2. The focus on worst case error is useful.

Weaknesses:
1. First and foremost this paper is missing a substantial amount of detail that would be required to make it self contained and able to be built upon. While the proposed procedure is described on a high level, it is very unclear how these parameters, such as the ridge parameter is to be chosen in practice. We are also missing what assumptions the authors are making on the potential outcomes. In the appendix the authors assume a spherical distribution, is this the case throughout?

2. There is a lack of framing in the context of the larger causal literature on this area. There is a fairly long literature which ties balancing weights to minimax estimation, a discussion should be provided in the text which contextualizes this work in the broader literature, c.f.:
Bruns-Smith, D., & Feller, A. (2022). "Outcome Assumptions and Balancing Weights" (arXiv:2203.09557
Kallus, Nathan. "Generalized optimal matching methods for causal inference." Journal of Machine Learning Research 21.62 (2020): 1-54.

3. The authors motivate the use of the ellipsoid by noting that balancing weights struggle in high dimensions because overlap fails to hold. This is a good motivation, but I don't quite see why this problem also wouldn't arise in this setting unless $\lambda$ is substantial (and thus induces a non-trivial amount of bias). It would seem to be the case that the hyperplane will fail to intersect the ellipsoid in high dimensions for the same reasons the probability of the test point being contained in the convex hull of A tends to 0 in that setting.

4. I'm unclear on theorem 2: The proof bounds the MSE by the rank of $\Sigma$, however if $AA^\top$ is positive semi-definite then $r=T$ for all $\lambda > 0$, which would seem to render the bound vacuous? Perhaps I am missing something here that would avoid this in the proof derivation, but it is hard to tell because the authors have excluded $\lambda$ from the proof.

5. It would be very helpful if the authors provided much more simulation results to build intuition about the behavior of the algorithm, and shows both success and failure cases explicitly.

---

> ### Author Response · Authors · 2025-05-24
> **Response to R8ch (Part 1) [Reviews 1 & 2]**
>
> We thank the reviewer for finding our approach and applications interesting, and for finding the worst-case error estimation is useful.
>
> We copy-paste reviews and  divide them in order to respond point-by-point.
>
> **Review 1: First and foremost this paper is missing a substantial amount of detail that would be required to make it self contained and able to be built upon. While the proposed procedure is described on a high level, it is very unclear how these parameters, such as the ridge parameter is to be chosen in practice. We are also missing what assumptions the authors are making on the potential outcomes. In the appendix the authors assume a spherical distribution, is this the case throughout?**
>
> **1. it is very unclear how these parameters, such as the ridge parameter is to be chosen in practice.**
>
> We thank the reviewer for raising the need for additional details on hyperparameter tuning. In our method, the parameter $\lambda$ is added to the covariance matrix $AA^{\top}$ when constructing the ellipsoidal matrix $Q$ to ensure it is positive definite, in line with standard practices in ellipsoidal geometry (see the ellipsoid definition in Equation 1).
>
> _We add the following part into appendix:_
>
> "Based on the ellipsoid definition Eq. 1, the matrix $Q$ must be positive definite. To ensure that eigenvalues of $Q$ are strictly positive, we add a small perturbation $\lambda \in (0, 1]$ to the diagonal of $AA^{\top}$. We use cross-validation on the pre-intervention period to select $\lambda$ that minimizes the $\ell_2$-norm. Therefore, such covariance matrix $AA^{\top}$ is used to construct $Q$ and the ellipsoid based on the definition Eq. 1”
>
> **2. We are also missing what assumptions the authors are making on the potential outcomes.**
>
> Recall our method is grounded in the potential outcome framework (Rubin, 1974), where the treated unit has an unobserved counterfactual outcome in the post-intervention. We consider the geometric approach to construct controls and potential outcomes.
>
> _We add the following to the text (section 4.1) for additional elaboration:_
> "So, our main assumption is that the true signal $a_1$ lies within an ellipsoid shape (intersection of a hyperplane and a higher-dimensional ellipsoid). This acts as a shape constraint on the treated unit, asserting that its underlying structure is similar to that of the controls (within the ellipsoid and hyperplane). This aligns with smoothness and low-complexity assumptions common in matrix completion used in causal inference. Further, having $a_1$ lying in the intersection follows the minimax optimality definition in which we can estimate the counterfactual."
>
> **3. In the appendix the authors assume a spherical distribution, is this the case throughout?**
>
> No. We restrict to the spherical distribution in the appendix only, to provide a geometric proof of consistency and unbiasedness (based on Halteman, 1986) that is fairly succinct (the full proof for the ellipsoidal case is longer and less instructive, in our view). We clarify that this assumption is used to provide intuition via geometry, and we emphasize that the results for spherical distribution extends to ellipsoidal.
>
> **Review 2: There is a lack of framing in the context of the larger causal literature on this area. There is a fairly long literature which ties balancing weights to minimax estimation, a discussion should be provided in the text which contextualizes this work in the broader literature, c.f.: Bruns-Smith, D., & Feller, A. (2022). and Kallus, Nathan. "Generalized optimal matching methods for causal inference."**
>
> We thank the reviewer for suggesting literature to strengthen our discussion.
>
> _We add the following paragraph into the Discussion section in text._
>
> "Other than standard SC, our approach is related in spirit to recent works that consider minimax optimization by balancing weights. For example, Bruns-Smith et al., (2022) leverage balancing weights and convex duality to derive causal estimators that minimize the worst-case bias under outcome model assumptions. Similarly, Kallus (2020) introduces the generalized optimal matching (GOM) framework, which considers a broad class of balancing and matching methods to minimize worst-case loss over hypothesis spaces, e.g., reproducing kernel Hilbert spaces (RKHS), while controlling variance through regularization.
>
> In contrast to these balancing-weights-based methods, our method takes a geometric approach where the solution lies within a covariance-induced ellipsoid. Our formulation avoids dependence on explicit outcome function classes or kernel choices, and rather than constructing balancing weights explicitly, we recover the treated unit as the Chebyshev center of the feasible region. As a result, our approach is particularly suited to settings with limited pre-treatment data, limited overlap, or ill-posed weight construction. This offers a complementary perspective to balancing-weight and function-space methods."

---

> ### Author Response · Authors · 2025-05-24
> **Response to R8ch (Part 2) [Reviews 3 & 4]**
>
> **Review 3: The authors motivate the use of the ellipsoid by noting that balancing weights struggle in high dimensions because overlap fails to hold. This is a good motivation, but I don't quite see why this problem also wouldn't arise in this setting unless $\lambda$ is substantial (and thus induces a non-trivial amount of bias). It would seem to be the case that the hyperplane will fail to intersect the ellipsoid in high dimensions for the same reasons the probability of the test point being contained in the convex hull of A tends to 0 in that setting.**
>
> We thank the reviewer for this important observation. We agree that in high dimensions, overlap and extrapolation might become geometrically challenging. Indeed, balancing weight methods often fail if the treated unit lies outside the convex hull and interpolation becomes unstable.
>
> In our case, rather than relying on interpolation within the convex hull, we have this feasible region resulting from intersection. While it is true that in high dimensions the intersection can become small, however, our formulation ensures that the intersection is non-empty, by construction, and the solution is well-defined. Recall, the hyperplane is constructed from pre-intervention of controls, and the ellipsoid is the pre- and post-intervention of controls, therefore, in panel data setting, intersection is non-empty.
> Also, unlike the convex hull, given that the ellipsoid expands in direction of high variance, it is not restricted to positive convex combinations, it can extrapolate along principal directions.
>
> Furthermore, the presence of $\lambda$ is substantial: the ellipsoid contracts in low-variance direction but never collapses due to the addition of ridge term to the covariance.
>
> _We add this discussion to the “properties of estimators” section in the text._
>
> **Review 4: I'm unclear on theorem 2: The proof bounds the MSE by the rank of $\Sigma$, however if $AA^{\top}$is positive semi-definite then $r=T$ for all $\lambda$>0, which would seem to render the bound vacuous? Perhaps I am missing something here that would avoid this in the proof derivation, but it is hard to tell because the authors have excluded $\lambda$ from the proof.**
>
> We thank the reviewer for this insightful comment and for pointing out a missing clarification in our theoretical analysis. We agree that when regularizing our covariance matrix, $AA^{\top}+\lambda I$, to ensure it is positive-definite, it becomes technically a full-rank for any $\lambda >0$. However, our aim in the theorem was to consider an unregularized version of the covariance matrix $AA^{\top}$. This aligns with standard conventions in matrix estimation literature, where low-rank structure is exploited for recovery and the ridge term is used solely to stabilize the inversion. To avoid confusion, we revise our proof by emphasizing that the covariance matrix is unregularized, and such that the rank is $r < T$, which prevents the bound from being vacuous.
> In addition, we consider another proof for regularized covariance where $\lambda$ is explicitly considered in derivations. The MSE is bounded by the rank $\tilde{r}$ which is strictly less than $T$ for all $\lambda > 0$.
>
> _Please see Appendix 9.2 for full proof and derivations._

---

> ### Author Response · Authors · 2025-05-24
> **Response R8ch (Part 3) [Review 5 and remaining comments]**
>
> **Review 5: It would be very helpful if the authors provided much more simulation results to build intuition about the behavior of the algorithm, and shows both success and failure cases explicitly.**
>
> 5.1. We thank the reviewer for suggesting to add more simulation results. We consider two additional sets of simulations, both added to Appendix. First, we consider an empirical simulation of the Penn World Table Dataset, inspired by Arkhangelsky, et al. (2019). Such simulations consider correlations within the dataset, and correlations between treatment assignment and covariates/other effects. Results show that EOpR outperforms other baseliens even under highly-correlated data, e.g. with treatment assignment prone to bias. See below table (All results are based on 100 simulations. First number is RMSE and second number between brackets is the standard deviation). _In appendix 10.2_
>
> | RMSE (std) | Democracy |  | Education |  | Random |  |
> |---|---|:---:|---|:---:|---|:---:|
> | EOpR | **0.154** | (0.099) | **0.125** | (0.064) | **0.179** | (0.148) |
> | SC (Abadie et al.) | 0.162 | (0.091) | 0.157 | (0.096) | 1.339 | (0.997) |
> | RSC (Amjad et al.) | 0.175 | (0.133) | 0.132 | (0.072) | 0.202 | (0.166) |
> | DSC (Ferman et al.) | 0.448 | (0.227) | 0.434 | (0.210) | 1.266 | (0.868) |
> | SDID (Arkhangelsky et. al) | 0.793 | (0.473) | 1.142 | (0.393) | 0.730 | (0.486) |
>
> Second we consider non-smooth synthetic simulations with outliers, a variation from the original simulation we have. Results show that even with existing outliers, EOpR constructs signal vectors that are sufficient for estimation. _Experiment setup and results in Appendix 10.1.3_
>
> **5.2. shows both success and failure cases explicitly.**
>
> _We explicitly mention failure cases in Discussion section in text as following:_
> "EOpR shows outstanding performance under various settings, especially under limited data availability, short pre-treatment periods (See Table 1) or noisy settings (See Table 2). However, it fails under certain settings. For example, if the treated unit deviates largely from the controls trends, in which it is structurally divergent from the ellipsoid, then the ellipsoid might not contain the true trajectory (shown in the COVID-19 experiment when increased $T_0$ when treated unit starts to deviate from other controls). Also, EOpR fails when pre-intervention controls have a square-shape matrix, such as $N=T_0$. In this case, the matrix is square and invertible which would create an exact interpolation of the pre-treatment treated unit. However, it creates an unstable feasible set, such that the Chebyshev center becomes sensitive to noise."
>
> **Summary of Requested Changes:**
>
> **1. Please add details in (a) the method definition, (b) related work, (c) assumptions, and (d) the theoretical framing.**
> We added more text in the method definition, related work, assumptions and theoretical framing. To summarize additions: .
> 1. In section 1, we add a new paragraph “Why Ellipsoids?” to motivate our quadratic signal class approach.
> 2. In section 2, we discuss more related works such as (Bruns-Smith & Feller, 2022) and (Kallus, 2020) , then discussed results from Bruns-Smith et al. (2025).
> 3. In section 4.1, we add a paragraph to clarify our geometric assumptions.
> 4. In section 4.4, we emphasize and clarify theoretical properties of method and estimator.
> 5. In section 6, we add a lengthy and thorough discussion on our method, comparing with other baselines and highlighting strengths and limitations.
> 6. In the appendix, we clarify the hyperparameter tuning strategy and add more explanation on the Riesz representation theorem.
>
> **2. Please explicitly consider λ in derivations**
>
> We thank the reviewer for this suggestion. We include a new proof for the regularized covariance where $\lambda$ is explicitly considered in derivations. The MSE is bounded by the rank $\tilde{r}$ which is strictly less than T for all $\lambda > 0$. _Please see Appendix 9.2 for full proof and derivations._ We updated Theorem 1 with the new bound considering the regularized covariance.
>
> **3. Please provide empirical simulation that clearly outlines the performance of the proposed method.**
>
> We thank the reviewer for suggesting additional empirical simulation. In addition to the generating process simulations we have, we added two alternative non-smooth simulations. We also add empirical simulation of the Penn World Table dataset, based on Arkhangelsky, et al. (2019). _Results in appendix 10.2, EOpR outperforms other baselines._
>
> **Finally, it would be useful, though not strictly required, if the authors compared the proposed approach to an unconstrained balancing weight estimator with a ridge penalty ... .**
>
> We found the suggested reference useful in building our discussion. Using this reference and other suggested references, we compare between our method and other balancing-weight methods in text as mentioned in (Part 1) Review 2 above.

---

### Comment · Action_Editor_oRLD · 2025-05-26
**Discussion**

Dear reviewers,

The authors have posted detailed replies to your reviews. Please read the rebuttals carefully and engage in the discussion with the authors.

Best,
TMLR Action Editor

---

### Decision · Action_Editor_oRLD · 2025-07-15

**Recommendation:** Reject

**Audience:**

Yes

**Audience Explanation:**

Reviewers agreed that the scope of this paper generally falls inside the communities interests and the specific angles the authors have taken is likely of interest to (parts of) the TMLR audience.

**Claims And Evidence:**

No

**Claims Explanation:**

While the core ideas of the paper were not received badly by the reviewers, the main criticism is a lack of clarity and precision along various dimensions. Reviewers struggled at times with justifications and explanations of the assumptions introduced in the submission in that it is hard to judge the claims under this uncertainty around the assumptions. Moreover, the submission would further benefit from a stronger empirical evaluation to convincingly back the claims and from a general improvement of clarity around the wording, notation, and theoretical considerations (see detailed comments in the reviews).

**Resubmission Of Major Revision:**

The authors may consider submitting a major revision at a later time.

---

> ### Author Response · Authors · 2025-07-18
>
> We wanted to confirm that the reviewers and action editor were able to see our most recent revision and responses, given they were only uploaded yesterday.  We ask especially since the changes were aimed at some of the points listed above such as clarity, empirical evaluation, and assumptions.